# SPECULATIVE DECODING: LOSSLESS SPEEDUP OF AUTOREGRESSIVE TRANSLATION

## ABSTRACT

Different from some previous work accelerating autoregressive translation (AT) at the sacrifice of quality, we propose Speculative Decoding (SpecDec) – a novel decoding paradigm inspired by speculative execution in computer architecture, which combines respective advantages of AT and non-autoregressive translation (NAT) for lossless speedup of translation. At each decoding step, SpecDec first speculatively drafts (i.e. decodes) next $k$ tokens with an NAT model and then verifies them with an AT model, where only the drafted tokens passing the verification are accepted as decoded tokens for guaranteeing its translation result is exactly the same as AT. The collaboration of NAT drafting and AT verification leads to a much higher decoding speed without quality loss due to parallel computing enabled by speculative decoding.

We conduct experiments in 4 standard WMT translation benchmarks and confirm the vanilla SpecDec yields exactly the same results as AT greedy decoding with an around $3\times$ speedup, and that its variant (SpecDec++) with an advanced verification strategy not only outperforms AT greedy decoding, but also further improves the decoding speed, resulting in an around $5\times$ speedup over AT. Moreover, SpecDec can be easily generalized for speeding up other seq2seq tasks like Abstractive Summarization, and benefit more from stronger computing devices, demonstrating its potential to become a *de facto* decoding standard in the future for efficient and lossless seq2seq generation. We will release all our codes and checkpoints to facilitate reproducing our results.

## 1 INTRODUCTION

Since the Transformer (Vaswani et al., 2017) prevailed in Natural Language Processing (NLP), autoregressive decoding has become the *de facto* standard for neural machine translation (NMT) as well as other generation tasks, because it is easy to train and reliable to generate high-quality results. Despite its advantages, autoregressive translation (AT) has been widely blamed for its poor inference efficiency, motivating non-autoregressive translation (NAT). Unlike AT which sequentially decodes only one token at each iteration so that the next token prediction can condition on the previous decoding results, NAT decodes tokens in parallel without depending on the surface form of previous tokens, largely improving the inference efficiency.

Recent research in NAT mainly focuses on improving its translation quality to bridge the performance gap between NAT and AT (Gu et al., 2018; Qian et al., 2021; Geng et al., 2021; Savinov et al., 2021). Until now, however, NAT's performance is still less reliable than AT, as NAT is more difficult than AT given its unawareness of the conditional dependence of translated tokens.

Given AT's reliable generation results and NAT's high efficiency, we propose an approach named Speculative Decoding (SpecDec) to combine their advantages, inspired by speculative execution[1], to accelerate translation without quality loss compared with AT. SpecDec decomposes a decoding iteration into two substeps: *draft* and *verify*: At each iteration, SpecDec first speculatively *drafts* (i.e., decodes) a fixed number of tokens[2] in parallel through NAT; Then, the drafted tokens are *verified*

---

[1] Speculative execution is an optimization technique used in computer architecture where a system performs some task in advance to avoid delays that would have to be incurred by doing the task after it is known that it is required (`https://wikipedia.org/wiki/Speculative_execution`).

[2] We use "*a block of* drafted tokens" to denote them in the following parts of this paper.

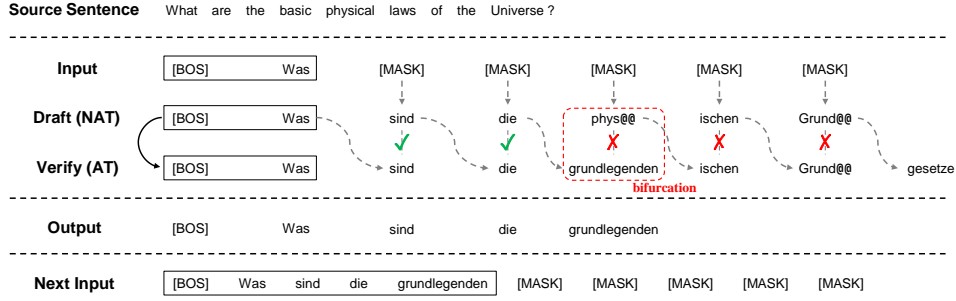

Figure 1: Speculative Decoding where a decoding iteration involves two substeps: *draft* and *verify*. In the **Draft** substep, an NAT model speculatively drafts (i.e., decodes) a block (block size $k = 5$ for this example) of tokens in parallel conditioning on the source sentence and previously decoded tokens (i.e., the tokens in the rectangle boxes). In the **Verify** substep, drafted tokens are verified in parallel: bifurcation is detected as the first position where we find the drafted token does not match the top-1 result verified by an AT model. The drafted tokens after the bifurcation position are all discarded, for guaranteeing SpecDec's translation is exactly the same with greedy decoding of AT.

by an AT model in an autoregressive manner to determine how many of them match AT's (top-1) results and thus can be accepted as translation results, as Figure 1 shows. In contrast to conventional AT which decodes at a low speed, AT verification is highly efficient because it performs in parallel; more importantly, it helps guarantee SpecDec's translation is identical to AT, resulting in a desirable balance between translation speed and quality, as shown in Figure 2.

In addition to the vanilla SpecDec whose translation is required (strictly by the top-1 matching criterion in AT verification) to be identical to greedy decoding of AT, we propose SpecDec++ — an advanced variant of SpecDec by slightly relaxing the rigid requirement during AT verification. SpecDec++ not only yields translations beyond greedy decoding, but also prevents good drafted tokens from being discarded just because they are different from greedy decoding results, leading to a higher inference speedup.

The experiments in four standard WMT benchmarks show that SpecDec can yield exactly the same translations as greedy decoding of AT with a $3\times$ speedup and that its variant SpecDec++ can outperform greedy decoding with an even higher ($\sim 5\times$) speedup. Moreover, the SpecDec paradigm can be easily generalized to other seq2seq tasks like Abstractive Summarization and benefit more from stronger computing devices. Its lossless quality and promising speedup results demonstrate its great potential to evolve into a *de facto* decoding standard for efficient seq2eq generation in the future.

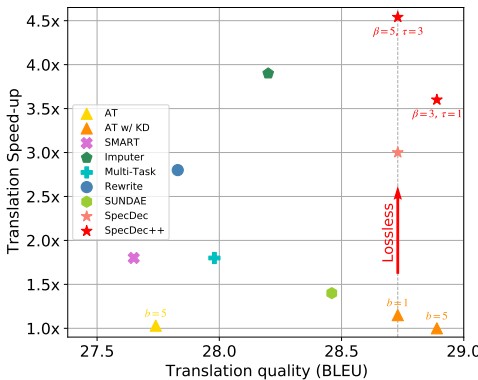

Figure 2: Translation quality and efficiency of models on WMT14 EN-DE. The speedup baseline ($1.0\times$) is the Transformer-base (Vaswani et al., 2017) with beam search. All models above except "AT" are trained with KD by a Transformer-big teacher.

## 2 BACKGROUND

### 2.1 AUTOREGRESSIVE TRANSLATION

Given a source sentence $\boldsymbol{x} = (x_1, x_2, \ldots, x_n)$ and the target sentence $\boldsymbol{y} = (y_1, y_2, \ldots, y_m)$, an autoregressive translation (AT) model is trained with the target distribution of conditional probabilities based on the chain rule:

$$\mathcal{L}_{\text{AT}} = \log P(\boldsymbol{y} \mid \boldsymbol{x}; \boldsymbol{\theta}_{\text{AT}}) = \sum_{i=1}^{m} \log P\left(y_i \mid \boldsymbol{y}_{<i}, \boldsymbol{x}; \boldsymbol{\theta}_{\text{AT}}\right) \tag{1}$$

where $\boldsymbol{y}_{<i}$ denotes previous target tokens before the $i^{th}$ position. As Eq (1) shows, an AT model is trained via the teacher-forcing strategy that uses target tokens as previously decoded tokens, which performs efficiently as the probability $P\left(y_i \mid \boldsymbol{y}_{<i}, \boldsymbol{x}\right)$ at each iteration can be calculated in parallel.

During inference, an AT model sequentially predicts output tokens given preceding decoded tokens:

$$\hat{\boldsymbol{y}} = \arg\max_{\boldsymbol{y}^*} \sum_{j=1}^{m'} \log P\left(y_j^* \mid \boldsymbol{y}_{<j}^*, \boldsymbol{x}; \boldsymbol{\theta}_{\text{AT}}\right) \tag{2}$$

where $\hat{\boldsymbol{y}} = (\hat{y}_1, \hat{y}_2, \ldots, \hat{y}_{m'})$ is the predicted output sentence.

Although AT offers desirable translation quality, its sequential decoding scheme with limited parallelism largely reduces its decoding speed, being its main efficiency bottleneck.

## 2.2 Non-Autoregressive Translation

To accelerate inference, non-autoregressive translation (NAT) (Gu et al., 2018) removes sequential dependence between target tokens with a conditional independence assumption:

$$\mathcal{L}_{\text{NAT}} = \sum_{i=1}^{m} \log P\left(y_i \mid \boldsymbol{x}; \boldsymbol{\theta}_{\text{NAT}}\right) \tag{3}$$

In contrast to AT that will not start predicting $y_j$ until $\boldsymbol{y}_{<j}$ are completely decoded, NAT decodes[3] the output sentence in parallel, which is much more efficient than AT:

$$\widetilde{\boldsymbol{y}} = \arg\max_{\boldsymbol{y}^*} \sum_{j=1}^{m'} \log P\left(y_j^* \mid \boldsymbol{x}; \boldsymbol{\theta}_{\text{NAT}}\right) \tag{4}$$

On the other hand, however, the conditional independence assumption makes it hard to train an NAT model well, leading to degradation in translation quality despite an improvement in decoding speed.

## 3 Speculative Decoding

Given the fact that AT translates better whereas NAT performs faster, we propose Speculative Decoding (SpecDec) to combine their respective advantages, inspired by speculative execution, to achieve lossless acceleration for seq2seq generation. Specifically, SpecDec decomposes every decoding iteration into two substeps – *draft* and *verify*:

**Draft**  At each iteration, SpecDec first utilizes an NAT model to simultaneously decode a block of drafted tokens (denoted as [MASK] in its decoder input in Figure 1) speculatively, conditioning on preceding translated tokens. Formally, given the source sentence $\boldsymbol{x} = (x_1, x_2, \ldots, x_n)$ and the previously translated tokens $\boldsymbol{y}_{\leq j} = (y_1, y_2, \ldots, y_j)$, SpecDec decodes the next $k$ (drafted) tokens as a block in parallel:

$$\widetilde{\boldsymbol{y}}_{j+1\cdots j+k} = \arg\max_{\widetilde{\boldsymbol{y}}_{j+1\cdots j+k}} \sum_{i=1}^{k} \log P\left(\widetilde{y}_{j+i} \mid \boldsymbol{y}_{\leq j}, \boldsymbol{x}; \boldsymbol{\theta}_{\text{NAT}}\right)$$

**Verify**  Then, the drafted tokens $\widetilde{\boldsymbol{y}}_{j+1\cdots j+k}$ are verified with an AT model in the autoregressive manner, which performs in parallel. We find the bifurcation position $c$ by comparing the drafted tokens with the autoregressive decoding results conditioning on the draft as Figure 1 shows:

$$c = \arg\max_i \frac{\mathbb{1}\left(\widetilde{y}_{j+i} \neq \hat{y}_{j+i}\right)}{i}, 1 \leq i \leq k$$
$$\hat{y}_{j+i} = \arg\max_{\hat{y}_{j+i}} \log P(\hat{y}_{j+i} \mid \boldsymbol{y}_{\leq j}, \widetilde{\boldsymbol{y}}_{j+1\cdots j+i-1}, \boldsymbol{x}; \boldsymbol{\theta}_{\text{AT}}) \tag{5}$$

where $\mathbb{1}(\cdot)$ is the indicator function and $\hat{y}_{j+i}$ is the top-1 result verified by the AT model conditioning on the previously translated tokens $\boldsymbol{y}_{<j}$ and the drafted tokens $\widetilde{\boldsymbol{y}}_{j+1\cdots j+i-1}$. We only accept the verified tokens before (including) the bifurcation position as translated tokens, which ensures that SpecDec yields the same results as greedy decoding of AT:

$$\boldsymbol{y}_{j+1\cdots j+c} = \hat{\boldsymbol{y}}_{j+1\cdots j+c} = (\widetilde{\boldsymbol{y}}_{j+1\cdots j+c-1}, \hat{y}_{j+c})$$

We iterate decoding with the above substeps until the termination condition is met, i.e. the [EOS] token is decoded or the sentence reaches the maximal length. As illustrated, SpecDec is highly efficient because both *draft* and *verify* perform in parallel.

---

[3]We use $\widetilde{\boldsymbol{y}}$ to denote NAT's translations, while we use $\hat{\boldsymbol{y}}$ to denote AT decoded/verified translation results.

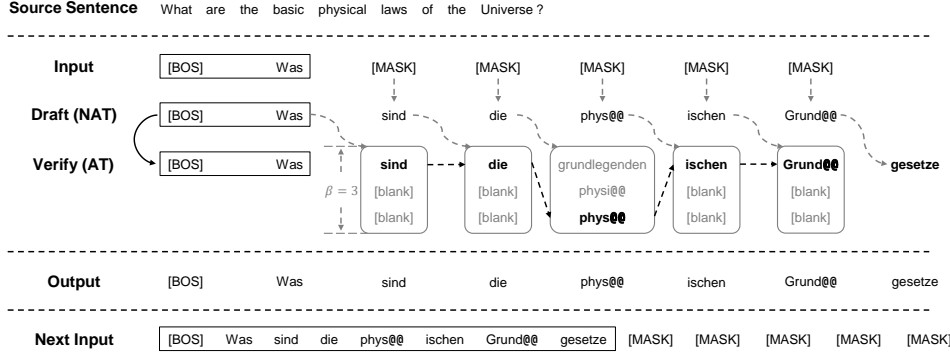

Figure 3: Illustration of SpecDec++. Compared to the vanilla SpecDec strictly requiring the drafted tokens to match the top-1 result of the AT verifier, SpecDec++ slightly relaxes the criterion to trust NAT's draft more, by only requiring the drafted tokens to fall in the *top-$\beta$* candidates of the AT verifier with a tolerable log-likelihood gap (not shown in this Figure; see Eq (7)). As a result, SpecDec++ allows more drafted tokens to be accepted even if they are slightly different from the top-1 result of the AT verifier, leading to a higher inference speedup.

## 3.1 NAT DRAFTER

As demonstrated above, an NAT model is the key to speculative decoding, which can efficiently generate a block of drafted tokens in parallel. Our NAT drafter differs from other NAT models in two aspects: First, we only require the NAT drafter to decode a block (i.e., fixed length) of tokens in each decoding iteration, instead of the whole sequence; Second, as illustrated in Figure 1, since we decode from *Left* to *Right*, the NAT drafter is required to decode tokens conditioning on the previously decoded tokens. Formally, given the source sentence $\boldsymbol{x} = (x_1, \cdots, x_n)$ and the randomly sampled prefix $\boldsymbol{y}_{\leq p}$ ($0 \leq p < m$) of the target sentence $\boldsymbol{y} = (y_1, \cdots, y_m)$, the model is trained to predict the next $k$ tokens, as shown in Figure 1:

$$\mathcal{L}_{\text{NAT}} = \sum_{i=p+1}^{p+k} \log P\left(y_i \mid \boldsymbol{y}_{\leq p}, \boldsymbol{x}; \boldsymbol{\theta}_{\text{NAT}}\right)$$

In addition, we leverage the glancing strategy following Qian et al. (2021), which exploits curriculum learning during training to get better performance. As in previous NAT work, we apply sequence-level knowledge distillation (Seq-KD) (Kim & Rush, 2016) by an autoregressive Transformer teacher model to train our NAT drafter.

## 3.2 AT VERIFIER

We use the conventional Transformer (see Section 2.1) as our AT verifier, which is the key to guaranteeing translation quality. As we hope as many drafted tokens by the NAT model as possible can be accepted by the AT verifier for a higher speedup, we also apply Seq-KD to the AT verifier by a shared teacher (with the NAT drafter), which not only allows the NAT drafter and AT verifier to perform similarly, but also improves the AT verifier's translation quality (Furlanello et al., 2018).

## 4 SPECDEC++

As shown in Figure 1 and discussed in Section 3, the vanilla SpecDec only accepts the drafted tokens that match the top-1 result of the AT verifier, which guarantees that SpecDec's translation is identical to greedy decoding of AT. However, the top-1 results are not necessarily better than the drafted tokens. As a result, the strict verification criterion (i.e., top-1 matching) will result in many good drafted tokens being discarded just because they are different from the top-1 result of the AT verifier, which limits the speedup of SpecDec.

To overcome this limitation, we propose a variant of SpecDec named SpecDec++, which is illustrated in Figure 3. Instead of the rigid top-1 matching requirement shown in Eq (5), SpecDec++ relaxes the criterion to trust NAT's draft more, by only requiring the drafted tokens to fall in *top-$\beta$* candidates with a tolerable (log-likelihood) score gap $\tau$ (away from the top-1 result):

$$\hat{y}_{j+i} = \begin{cases} \widetilde{y}_{j+i}, \text{ if } \textit{SpecDec++ requirement} \text{ is met} \\ \text{same as Eq (5), otherwise} \end{cases}$$

As discussed above, *SpecDec++ requirement* is met if Eq (6) and (7) are both true:

$$\log P(\widetilde{y}_{j+i} \mid \boldsymbol{y}_{\leq j}, \boldsymbol{x}; \boldsymbol{\theta}_{\text{NAT}}) \geq \log P(\hat{y}_{j+i}^{(\beta)} \mid \boldsymbol{y}_{\leq j}, \widetilde{\boldsymbol{y}}_{j+1\cdots j+i-1}, \boldsymbol{x}; \boldsymbol{\theta}_{\text{AT}}) \tag{6}$$

$$\log P(\hat{y}_{j+i}^{(1)} \mid \boldsymbol{y}_{\leq j}, \widetilde{\boldsymbol{y}}_{j+1\cdots j+i-1}, \boldsymbol{x}; \boldsymbol{\theta}_{\text{AT}}) - \log P(\widetilde{y}_{j+i} \mid \boldsymbol{y}_{\leq j}, \boldsymbol{x}; \boldsymbol{\theta}_{\text{NAT}}) \leq \tau \tag{7}$$

where $\log P(\hat{y}_{j+i}^{(\beta)} \mid \cdot)$ is the top-$\beta$ ranked result's log-likelihood score by the AT verifier.

The advanced verification criterion with the hyperparameter top-$\beta$ and tolerance $\tau$ not only allows more drafted tokens to be accepted for a higher speedup but also enables SpecDec++ to yield translations beyond greedy decoding.

## 5 EXPERIMENTS

### 5.1 EXPERIMENTAL SETTINGS

**Datasets and Evaluation**   We mainly evaluate our approach on the most recognized machine translation benchmark: WMT14 English-German translation which contains 4.5M translation pairs for training. Following prior work (Ott et al., 2018), we adopt *newstest-13* as our validation set for finding the best hyperparameters and model checkpoints, and test on *newstest-14*. We use 32K Byte Pair Encoding (BPE) (Sennrich et al., 2016) subwords[4] as the joint source-target dictionary. Following prior work in NAT, we report BLEU (Papineni et al., 2002) to facilitate translation quality comparison. For inference efficiency, we use both the average number of decoding iterations and speedup over beam search. Specifically, we test the inference speed by running the model with one sentence at a time (batch=1)[5] using fairseq implementation[6] on 1 Nvidia P100 GPU.

In addition to WMT14 EN-DE, we also test SpecDec on WMT14 DE-EN and WMT16 EN-RO/RO-EN benchmarks, as previous NAT work.

**Model Configuration**   We mainly conduct experiments on the most commonly used base-size Transformer (Vaswani et al., 2017) architecture. The Transformer-base[7] has a 6-layer encoder and a 6-layer decoder. Its embedding/FFN dimension/#heads are 512/2,048/8. We use the model architecture for both the drafter (NAT) and the verifier (AT). We apply sequence-level knowledge distillation as discussed in Section 3.1 and 3.2 for both the drafter and verifier using a shared teacher. Following recent iterative NAT work (Saharia et al., 2020; Savinov et al., 2021), we use the Transformer-big as the teacher for WMT14 EN-DE/DE-EN[8] and use Transformer-base for WMT16 EN-RO/RO-EN, which all train with the raw training set and generate the distilled training set with beam search ($b = 5$). We include model training details in Appendix A.

### 5.2 MAIN RESULTS

The translation quality and speedup results on WMT14 EN-DE are presented in Table 1. Unlike previous NAT approaches that are inferior to AT with Seq-KD (i.e., our AT verifier), SpecDec introduces an around $3\times$ speedup with exactly the same translation quality as (autoregressive) greedy decoding by our AT verifier, truly achieving lossless acceleration. SpecDec++ further improves the results by relaxing the strict top-1 matching criterion: slightly relaxing (i.e., SpecDec++ high-quality) allows us to achieve better translation than greedy decoding – even approaching the beam search result with a higher speedup ($3.0\times \rightarrow 3.6\times$), and a little more aggressively relaxing (i.e., SpecDec++ high-efficiency) further accelerates inference ($3.6\times \rightarrow 4.5\times$) owing to the acceptance of more tokens despite a marginal loss of translation quality.

---

[4]We use the same BPE tokenization and vocabulary as Ghazvininejad et al. (2019).

[5]We report performances with various batch sizes in Appendix E.

[6]https://github.com/pytorch/fairseq. Beam search ($b = 5$) is our speed baseline ($1.0\times$).

[7]We also include results of a Transformer with a deep encoder and a shallow decoder in Appendix B.5.

[8]As a reference, the teacher models of Saharia et al. (2020) achieve 29.5 and 32.2 BLEU in WMT14 EN-DE and DE-EN respectively, which are stronger than our teachers achieving 29.3 and 32.1 respectively.

| | Models | Iter. | Speedup | | WMT14 | | WMT16 | |
|---|---|---|---|---|---|---|---|---|
| | | | | | EN-DE | DE-EN | EN-RO | RO-EN |
| AT baseline | Transformer-base (w/o Seq-KD)[†] | N | - | 1.0× | 27.74 | 31.09 | 34.28 | 33.99 |
| Fully NAT | NAT w/ Fertility (Gu et al., 2018) | 1 | 15.6× | - | 17.69 | 21.47 | 27.29 | 29.06 |
| | CTC (Libovický & Helcl, 2018) | 1 | - | - | 17.68 | 19.80 | 19.93 | 24.71 |
| | AXE (Ghazvininejad et al., 2020a)[*] | 1 | - | - | 23.53 | 27.90 | 30.75 | 31.54 |
| | GLAT (Qian et al., 2021) | 1 | 15.3× | - | 25.21 | 29.84 | 31.19 | 32.04 |
| | OaXE (Du et al., 2021)[*] | 1 | - | - | 26.10 | 30.20 | 32.40 | 33.30 |
| | AligNART (Song et al., 2021) | 1 | 13.4× | - | 26.40 | 30.40 | 32.50 | 33.10 |
| | DSLP (Huang et al., 2021) | 1 | 14.8× | - | 27.02 | **31.61** | **34.17** | **34.60** |
| | F-VAE (Gu & Kong, 2021) | 1 | 16.5× | - | 27.49 | 31.10 | 33.79 | 33.87 |
| | REDER (Zheng et al., 2021) | 1 | 5.5× | - | **27.50** | 31.25 | 33.60 | 34.03 |
| | DA-Transformer (Huang et al., 2022a)[*] | 1 | 13.9× | - | 27.49 | 31.37 | - | - |
| Iterative NAT | iNAT (Lee et al., 2018) | 10 | - | - | 21.61 | 25.48 | 29.32 | 30.19 |
| | CMLM (Ghazvininejad et al., 2019)[*] | 10 | 1.7× | 1.8× | 27.03 | 30.53 | 33.08 | 33.31 |
| | LevT (Gu et al., 2019) | 2.1 | 4.0× | 3.7× | 27.27 | - | - | 33.26 |
| | SMART (Ghazvininejad et al., 2020b)[*] | 10 | 1.7× | 1.8× | 27.65 | 31.27 | - | - |
| | DisCo (Kasai et al., 2020)[*] | 4.8 | 3.5× | - | 27.34 | 31.31 | 33.22 | 33.25 |
| | Imputer (Saharia et al., 2020)[*] | 8 | 3.9× | - | 28.20 | 31.80 | 34.40 | 34.10 |
| | Multi-Task NAT (Hao et al., 2021)[*] | 10 | 1.7× | 1.8× | 27.98 | 31.27 | 33.80 | 33.60 |
| | RewriteNAT (Geng et al., 2021)[*] | 2.7 | 3.1× | 2.8× | 27.83 | 31.52 | 33.63 | 34.09 |
| | SUNDAE (Savinov et al., 2021)[*] | 16 | 1.4× | - | 28.46 | 32.30 | - | - |
| | CMLMC (Huang et al., 2022b)[*] | 10 | - | 1.5× | 28.37 | 31.41 | 34.57 | 34.13 |
| Ours | Teacher | N | - | / | 29.31[‡] | 32.11[‡] | 34.72 | 34.49 |
| | NAT drafter (k = 25) | 1.6 | - | 14.3× | 26.48 | 30.23 | 32.08 | 32.21 |
| | AT verifier (b = 5) | N | - | 1.0× | 28.89 | 32.53 | 34.96 | 34.86 |
| | AT verifier (b = 1) | N | - | 1.1× | 28.73 | 32.18 | 34.83 | 34.65 |
| | SpecDec (k = 25) | 4.9 | - | 3.0× | 28.73 | 32.18 | 34.83 | 34.65 |
| | SpecDec++ (k = 25, high-quality) | 4.0 | - | 3.6× | **28.89** | **32.56** | **35.32** | **34.98** |
| | SpecDec++ (k = 25, high-efficiency) | 3.1 | - | **4.5×** | 28.73 | 32.19 | 34.92 | 34.80 |

Table 1: Results of SpecDec on four standard WMT benchmarks. For inference efficiency, we report the averaged decoding iteration (denoted as *Iter.*) and speedup on WMT14 EN-DE. The results that truly match or outperform autoregressive decoding (i.e., AT verifier with $b = 1$) are highlighted in red. [†] AT baselines' results are from Ghazvininejad et al. (2019). [‡] indicates results obtained by Transformer-big. [*] denotes models distilled by a Transformer-big model. While speedup numbers reported by the previous papers (left column) are not strictly comparable to ours due to different devices and environments, we reran some of them that are open source using the same device and environment as ours and report their speedups (right column), observing no significant difference.

By looking into the results, our NAT drafter's translation quality is better than the majority of early fully NAT work but inferior to most iterative NAT approaches. Compared with the NAT models including complicated mechanisms such as length prediction, length beam, reranking, and CTC that slow down the efficiency per iteration[9], our NAT drafter is simple and straightforward. As a result, its decoding efficiency per iteration is higher – even comparable to fully NAT despite taking 1.6 decoding iterations on average. The acceptable translation quality and high efficiency of our NAT drafter significantly help accelerate autoregressive decoding, playing an instrumental role in lossless acceleration of SpecDec.

Results for other language pairs are similar to WMT14 EN-DE. We include details in Table 8 in Appendix B.1.

## 5.3 ANALYSIS

### 5.3.1 HYPERPARAMETER

**Block Size** $k$    We conduct experiments with various block sizes on the development set and show the results in Table 2. As the block size $k$ increases, the number of mean accepted tokens, which highly correlates with speedup and the number of decoding iterations, first increases and reaches a peak when $k = 25$. Further increasing $k$ has an adverse effect, because it will become very hard for the model to learn to translate too many tokens simultaneously given the limited model capacity, leading to a drop in both efficiency and quality.

**Top-$\beta$ and Tolerance $\tau$ in SpecDec++**    We study the effects of hyperparameters in SpecDec++: top-$\beta$ and tolerance $\tau$, and show the results on the development set in Table 3. Moderately increasing $\tau$ and $\beta$ not only leads to an increase of mean accepted tokens since AT verification becomes

---

[9]For example, RewriteNAT (Geng et al., 2021) uses length beam size 5, slowing down its efficiency.

| Models | $k$ | Tok. | BLEU | Speed |
|---|---|---|---|---|
| AT ($b = 5$) | - | 1.00 | 26.72 | 1.00× |
| SpecDec++ | 10 | 5.97 | 26.68 | 3.04× |
| | 15 | 6.74 | **26.94** | 3.47× |
| | 20 | 7.24 | 26.75 | 3.55× |
| | 25 | **7.56** | 26.92 | **3.79×** |
| | 30 | 7.44 | 26.75 | 3.63× |

Table 2: The mean accepted tokens (*Tok.*), the translation quality (*BLEU*), and the efficiency (*Speed*) when decoding with a various number of block size $k$ on the development set (*newstest-13*). The results are obtained with SpecDec++ (*top-3, $\tau = 1.0$*).

| Models | $\tau$ | Top-3 ($\beta = 3$) | Top-5 ($\beta = 5$) |
|---|---|---|---|
| SpecDec++ | 1 | 7.56/**27.02** | 7.58/27.02 |
| | 2 | 8.64/26.92 | 8.77/26.92 |
| | 3 | 9.46/26.84 | 9.72/26.84 |
| | 4 | 10.04/26.78 | 10.50/26.74 |
| | 5 | 10.38/26.70 | **10.99**/26.64 |

Table 3: Results on the development set (*newstest-13*) with different hyperparameters in SpecDec++ ($k = 25$). Each cell lists the mean accepted tokens and BLEU score. The BLEU score of greedy decoding of the AT verifier is 26.62.

| | Models | Iteration | BLEU | Speed |
|---|---|---|---|---|
| Blockwise Stern et al. (2018) | Blockwise decoding ($k = 1$) | N | 29.11 | 1.0×[†] |
| | Blockwise decoding ($k = 2$) | / | 28.95 | 1.7×[†] |
| | Blockwise decoding ($k = 10$) | / | 27.40 | 3.0×[†] |
| Ours | Teacher (Transformer-big, $b = 5$) | N | 29.31 | 1.0× |
| | AT verifier ($b = 5$) | N | 29.25 | 1.0× |
| | AT verifier ($b = 1$) | N | 29.18 | 1.1× |
| | NAT drafter ($k = 10$) | 3.3 | 27.41 | 7.2× |
| | SpecDec ($k = 10$) | 5.4 | 29.18 | 2.7× |
| | SpecDec++ ($k = 10$, *top-3, $\tau = 1.0$*) | 5.0 | 29.28 | 2.9× |
| | SpecDec++ ($k = 10$, *top-5, $\tau = 6.0$*) | 3.9 | 29.12 | 3.8× |
| | NAT drafter ($k = 30$) | 1.4 | 27.35 | 15.0× |
| | SpecDec ($k = 30$) | 4.8 | 29.18 | 3.0× |
| | SpecDec++ ($k = 30$, *top-3, $\tau = 1.0$*) | 4.2 | **29.32** | 3.5× |
| | SpecDec++ ($k = 30$, *top-5, $\tau = 6.0$*) | 2.6 | 29.15 | **5.0×** |

Table 4: Results of SpecDec of the big-size model configuration on WMT14 EN-DE and the comparison to the state-of-the-art Blockwise Decoding Stern et al. (2018). [†] denotes the speedup results reported in original papers obtained by comparison with greedy decoding.

less strict but also improves the translation quality over greedy decoding. However, the translation quality will decrease if the constraints are over relaxed: the BLEU score will degrade from the peak of 27.02 to 26.64 when decoding with *top-5* selection (i.e., $\beta = 5$) and $\tau = 5.0$. Based on the results in the development set, we conservatively select $\beta = 3, \tau = 1.0$ for the *high-quality* SpecDec++, and use $\beta = 5, \tau = 3.0$ as the *high-efficiency* SpecDec++ to pursue the higher speedup without substantial loss of translation quality for WMT14 EN-DE as in Table 1.

### 5.3.2 MODEL SIZE

In addition to models of the base-size configuration, we also study larger models to test the effectiveness of SpecDec. We here use Transformer-big (Vaswani et al., 2017) as our model architecture for both the NAT drafter and the AT verifier in SpecDec/SpecDec++[10], and compare it with the conventional Transformer-big baseline as well as Blockwise Decoding (Stern et al., 2018) – a state-of-the-art efficient Transformer-big variant by introducing additional $k - 1$ heads on top of the Transformer decoder to generate next $k$ tokens as a block and verifies, which works in a similar way to ours. According to Table 4, our SpecDec/SpecDec++ substantially speeds up the AT baselines and outperforms Blockwise Decoding with both better results and a higher speedup. Compared with Blockwise Decoding whose performance drops significantly when $k$ increases to 10 due to its underinvestment in speculation by only using lightweight heads to generate the next few tokens in parallel, SpecDec/SpecDec++ using an independent NAT drafter is much more powerful to generate more drafted tokens that can be accepted, turning out to result in a significantly higher speedup, despite introducing more parameters (see Appendix C for detailed discussion of memory cost issues).

---

[10]The hyperparameters (e.g., block size $k$, Top-$\beta$, tolerance $\tau$) in SpecDec (big) are re-tuned on the development set, which may be different from those in the base-size models.

| Models | Drafter | Tok. | Iter. | BLEU |
|---|---|---|---|---|
| AT-base (greedy) | - | 1.00 | N | 28.73 |
| SpecDec-base | lightweight heads | 2.23 | 11.5 | 28.73 |
| | NAT-base | 5.53 | 5.0 | 28.73 |
| | NAT-big | **5.90** | **4.7** | **28.73** |
| SpecDec++-base (*top-3*, $\tau = 1.0$) | lightweight heads | 2.35 | 10.9 | 28.11 |
| | NAT-base | 6.69 | 4.1 | 28.81 |
| | NAT-big | **7.32** | **3.8** | **29.12** |
| SpecDec++-base (*top-5*, $\tau = 3.0$) | lightweight heads | 2.61 | 10.6 | 27.02 |
| | NAT-base | 8.71 | 3.2 | 28.58 |
| | NAT-big | **9.73** | **2.8** | **28.98** |

Table 5: Results of SpecDec-base ($k = 30$) with various sizes of drafters on WMT14 EN-DE.

| | Models | Iteration | R-1 | R-2 | R-L | Speed |
|---|---|---|---|---|---|---|
| Teacher | BART (Lewis et al., 2020) | N | 44.16 | 21.28 | 40.90 | - |
| AT (w/o Seq-KD) | BART-base ($b = 5$) | N | 42.84 | 20.08 | 39.51 | 1.0× |
| Ours | NAT drafter ($k = 25$) | 3.3 | 37.10 | 14.87 | 33.47 | 15.0× |
| | AT verifier ($b = 5$) | N | 42.55 | 19.83 | 39.31 | 1.0× |
| | AT verifier ($b = 1$) | N | 43.00 | 20.28 | 39.96 | 1.1× |
| | SpecDec ($k = 25$) | 14.0 | **43.00** | **20.28** | **39.96** | 3.0× |
| | SpecDec++ ($k = 25$, *top-3*, $\tau = 1.0$) | 10.8 | 42.95 | 20.24 | 39.73 | 3.7× |
| | SpecDec++ ($k = 25$, *top-5*, $\tau = 3.0$) | 7.9 | 42.02 | 19.42 | 38.66 | **4.8×** |

Table 6: Results of SpecDec-base model on CNN-DM for Abstractive Summarization.

Moreover, we observe that big-size models can use a larger block size ($k = 30$) than the base-size models ($k = 25$) since larger capacity equips the model with a more powerful ability to learn to decode more tokens well in parallel. To better demonstrate this point, we conduct a comparative study of using drafters of different sizes in SpecDec-base, given the same block size ($k = 30$). According to Table 5, the big-size NAT drafter largely outperforms the base-size counterpart and the drafter with lightweight heads (as used in Stern et al. (2018)) performs worst, demonstrating that a stronger drafter can generate drafted tokens more reliably (i.e., on average more drafted tokens accepted by the AT verifier), resulting in fewer decoding iterations, which indicates that SpecDec may be further improved if a more powerful (not necessarily larger) NAT drafter is equipped.

### 5.3.3 OTHER SEQ2SEQ TASKS

We test SpecDec's effectiveness in one of the most representative seq2seq tasks – Abstractive Summarization. We employ the distilled training data of CNN Daily Mail (Hermann et al., 2015) from BART (Lewis et al., 2020) to train the NAT drafter and the AT verifier whose model architectures are both BART-base, and test on the CNN Daily Mail test split following previous work.

According to Table 6, the vanilla SpecDec consistently achieves exactly the same result as the AT verifier ($b = 1$), which is, to the best of our knowledge, the first work that achieves such a 3× lossless speedup for Abstractive Summarization. SpecDec++ further accelerates inference but does not show any quality improvement as observed in NMT experiments because of the larger performance gap between the NAT drafter and the AT verifier in the abstractive summarization benchmark.

### 5.4 DISCUSSION

Extensive experiments in multiple tasks show that SpecDec can significantly speed up seq2seq generation without quality loss. The state-of-the-art lossless speedup is attributed to the substantially improved computational parallelism that allows better utilization of (hardware) computing resources. We believe SpecDec is promising and can even benefit more from evolving processor hardware that will become increasingly powerful and better at parallel computing (shown in Appendix E).

As a preliminary study, SpecDec is far from perfect and has much headroom for improvement. First, according to the experimental results above, we know SpecDec's translation quality mainly depends on the AT verifier and its efficiency relies on the NAT drafter (whose capability matters how many drafted tokens can be accepted). We believe more powerful NAT/AT models (than the simple and naive ones used in this paper) will benefit SpecDec to achieve better results.

Moreover, SpecDec's potential can be further exploited by optimizing its implementation in computing and memory access. For example, according to Table 18 showing time cost by modules in SpecDec++, our naive implementation costs a total of approximately 16% of overall time to (sequentially) encode the input for AT and NAT. Obviously, this part can be optimized by performing AT and NAT encoding in parallel because they are independent, or sharing (or partially sharing) AT's encoder with NAT, which we leave as future exploration. Also, the NAT decoder costs more than the AT decoder because it employs bi-directional attention and cannot save the computation for the already decoded tokens as AT, which we believe can be improved in the future with a better non-autoregressive decoding mechanism designed for SpecDec.

# 6 RELATED WORK

**Non-autoregressive Decoding** To accelerate autoregressive translation (AT), Gu et al. (2018) first proposed Non-Autoregressive Translation (NAT), which decodes the output sentence in one single iteration despite translation quality loss. Recent work mainly focused on improving the quality while maintaining competitive speedups, including applying various training objectives (Wang et al., 2019; Wei et al., 2019; Shu et al., 2020; Shao et al., 2020; Guo et al., 2020; Liu et al., 2021; Ding et al., 2021b; Zeng et al., 2022; Shao et al., 2022), modeling dependencies between target tokens (Kaiser et al., 2018; Sun et al., 2019; Ghazvininejad et al., 2019; Liu et al., 2020; Bao et al., 2021; Zhan et al., 2022; Zhu et al., 2022) and refining the translation outputs with multi-pass iterations (Chan et al., 2019; Stern et al., 2019; Sun & Yang, 2020; Ghazvininejad et al., 2020b; Ding et al., 2021a; Norouzi et al., 2022). However, due to the inherent conditional independence assumption, NAT models' translation quality is generally less reliable than AT.

**Semi-autoregressive Decoding** There are also some attempts trying to combine autoregressive (AR) and non-autoregressive (NAR) decoding: Wang et al. (2018) proposed to utilize NAR decoding locally while keeping the AR property globally; on the contrary, Ran et al. (2020) and Kong et al. (2020) introduced a local-AR model which retained the NAR property globally. Similar ideas have been also proposed for Grammatical Error Correction (GEC): Chen et al. (2020) proposed to use a sequence tagger to identify the grammatical errors' spans and then use AR decoding to correct them; Aggressive Decoding (Sun et al., 2021) is the first work that introduces speculative decoding into GEC. It assumes that the input is the sentence to be generated in the future (i.e., there are no grammatical errors in the input), and then verifies the whole sentence in parallel through greedy decoding of AT. However, Aggressive Decoding works only for tasks where the input and output are highly similar, which limits its application. The most similar work to ours is Blockwise Decoding (Stern et al., 2018) that proposed to additionally insert $k - 1$ NAR heads on top of the Transformer decoder to generate $k$ positions in parallel and use the original AR head to verify these outputs. However, its underinvestment in the NAR modeling seriously limits its performance, resulting in a much lower efficiency than our approach.

# 7 CONCLUSION AND FUTURE WORK

We propose a novel Speculative Decoding (SpecDec) paradigm as well as its variant (SpecDec++) by combining respective advantages of AT and NAT. Contrary to the stereotype that more models (parameters) tend to slow down inference, SpecDec's introduction of an additional NAT model substantially speeds up AT without quality loss, achieving a state-of-the-art lossless acceleration result owing to higher computational parallelism introduced by the idea of speculative execution to better utilize computing resources.

Besides machine translation, SpecDec can be easily generalized to other seq2seq tasks like Abstractive Summarization and benefit from stronger computing devices (to discuss in Appendix E). It demonstrates a novel yet promising perspective for efficient seq2seq generation, orthogonal to the efforts for advancing the state-of-the-art NAT and AT models that can further benefit SpecDec.

Despite the state-of-the-art results, SpecDec still has great potential with much headroom for improvement, as we discuss in Section 5.4. We hope that our preliminary study could draw more attention to improving this promising decoding paradigm with potential to evolve into a *de facto* standard for efficient and lossless seq2seq generation in the near future.

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

APPENDIX

## A  HYPERPARAMETERS

Hyper-parameters of training SpecDec are listed in Table 7. Following Vaswani et al. (2017) and Ott et al. (2018), we also average model parameters from the last 10 checkpoints.

| Hyperparameter | Value | Hyperparameter | Value |
|---|---|---|---|
| devices | 8 Nvidia V100 GPU | devices | 8 Nvidia V100 GPU |
| label smoothing | 0.1 | label smoothing | 0.1 |
| # max tokens | 20000 | # max tokens | 4096 |
| update frequency | 4 | update frequency | 4 |
| dropout rate | [0.1, 0.2, 0.3] | dropout rate | [0.1, 0.2, 0.3] |
| max source positions | 1000 | max source positions | 1000 |
| max target positions | 1000 | max target positions | 1000 |
| Adam lr | $1 \times 10^{-3}$ | Adam lr | $5 \times 10^{-4}$ |
| Adam $\beta_1$ | 0.9 | Adam $\beta_1$ | 0.9 |
| Adam $\beta_2$ | 0.99 | Adam $\beta_2$ | 0.999 |
| lr-scheduler | inverse square | Adam $\epsilon$ | $1 \times 10^{-6}$ |
| warm-up lr | $1 \times 10^{-7}$ | lr-scheduler | inverse square |
| weight decay | 0.00001 | warm-up lr | $1 \times 10^{-7}$ |
| clip norm | 3.0 | weight decay | 0.01 |
| # warmup updates | 4000 | clip norm | 5.0 |
| max updates | 100K | # warmup updates | 10000 |
| max epoch | 1000 | max updates | 300K |

Table 7: Hyper-parameters and settings of the AT verifier (left) and the NAT drafter (right).

## B  DETAILS OF EXPERIMENTAL RESULTS

### B.1  SPEEDUP RESULTS FOR MACHINE TRANSLATION

While we only report the speedup results of EN-DE translation in the main body of this paper due to space limitation, we here report speedup details of EN-DE, DE-EN, EN-RO and RO-EN in Table 8. As demonstrated in Section 5.3.1, we tune the hyperparameters ($k$, $\beta$ and $\tau$) of SpecDec/SpecDec++ on the development set for each direction of benchmarks. According to Table 8, SpecDec/SpecDec++ performs consistently well on all the language pairs despite slight differences in speedup ratios.

| Models | | EN-X | | | | X-EN | | | |
|---|---|---|---|---|---|---|---|---|---|
| | | $\beta, \tau$ | Iter. | BLEU | Speed | $\beta, \tau$ | Iter. | BLEU | Speed |
| AT | AT verifier ($b=1$) | - | N | 28.73 | 1.1× | - | N | 32.18 | 1.1× |
| | NAT drafter | - | 1.6 | 26.48 | 14.3× | - | 1.5 | 30.23 | 14.0× |
| WMT14 | SpecDec ($k = 25$) | - | 4.9 | 28.73 | 3.0× | - | 4.4 | 32.18 | 3.0× |
| EN-DE | SpecDec++ ($k = 25$, *high-quality*) | 3,1.0 | 4.0 | 28.89 | 3.6× | 3,1.0 | 3.4 | 32.56 | 3.9× |
| | SpecDec++ ($k = 25$, *high-efficiency*) | 5,3.0 | 3.1 | 28.73 | 4.5× | 5,3.0 | 2.5 | 32.19 | 4.8× |
| AT | AT verifier ($b=1$) | - | N | 34.83 | 1.1× | - | N | 34.65 | 1.1× |
| | NAT drafter | - | 1.6 | 32.08 | 13.3× | - | 1.6 | 32.21 | 13.8× |
| WMT16 | SpecDec ($k = 25$) | - | 6.4 | 34.83 | 2.2× | - | 5.9 | 34.65 | 2.4× |
| EN-RO | SpecDec++ ($k = 25$, *high-quality*) | 5,4.0 | 4.5 | 35.32 | 3.1× | 3,2.0 | 4.6 | 34.98 | 2.9× |
| | SpecDec++ ($k = 25$, *high-efficiency*) | 5,6.0 | 4.0 | 34.92 | 3.3× | 5,3.0 | 4.0 | 34.80 | 3.3× |

Table 8: Details of the main results of SpecDec/SpecDec++ on WMT benchmarks. *X* denotes the corresponding language in each benchmark (*German* in WMT14 EN-DE and *Romanian* in WMT16 EN-RO). $k$ is the block size. $\beta$ indicates the Top-$\beta$ selection in SpecDec++ and $\tau$ is the tolerance hyperparameter. All hyperparameters ($k$, $\beta$ and $\tau$) are tuned on the development set of each benchmark.

## B.2 SPEEDUP DISTRIBUTION

To further understand the acceleration effects of SpecDec, we present the speedup distribution of a single sentence on the WMT14 EN-DE test set (which has 3,003 sentences in total) in Figure 4, showing that most sentences are translated with a $3\times \sim 6\times$ speedup compared to the beam search baseline, while some rare cases can even achieve over $10\times$ speedup.

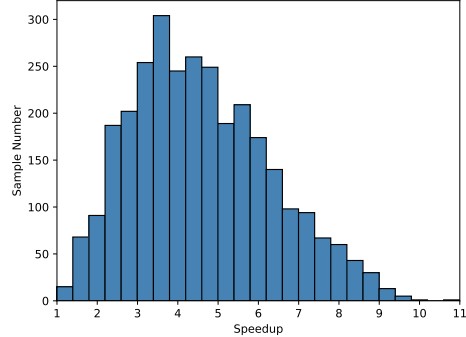

Figure 4: Single sentence speedup distribution by SpecDec++ ($k = 25$, *high-efficiency*).

| Models | BLEU | Rep. |
|---|---|---|
| AT (greedy) | 28.73 | 0.18% |
| CMLM ($T = 4$) | 25.75 | 1.13% |
| CMLM ($T = 10$) | 27.09 | 0.24% |
| SpecDec | 28.73 | 0.18% |
| SpecDec++ | **28.89** | **0.17%** |

Table 9: Token repetition ratio on WMT14 EN-DE. SpecDec-base is test with hyper-parameters $k = 25$, *top-3*, $\tau = 1.0$. CMLM is tested with our implementation with the length beam set to 3.

## B.3 WORD REPETITIONS

With the conditional independence assumption, NAT models show a serious weakness in modeling highly multimodal distributions. The token repetition ratio is often utilized as a proxy to measure this *multi-modality* problem, which represents the degree of the text inconsistency. However, the role of our AT verifier guarantees that this problem does not exist in SpecDec. As shown in Table 9, the token repetition ratio of SpecDec/SpecDec++ is similar to that of our AT baseline, which is significantly lower than most relevant NAT models.

## B.4 TEACHER MODEL(S)

We study the effects of the teacher[11] on SpecDec, by comparing the results of a single teacher with the teacher ensemble of 3 Transformer-big models in Table 10. Compared with a single teacher model, teacher ensemble improves all the NAT drafter, AT verifier, and end-to-end SpecDec/SpecDec++ results, indicating that our approach can benefit from a better teacher.

| Models | | Single Teacher | | | Teacher Ensemble | | |
|---|---|---|---|---|---|---|---|
| | | Iter. | BLEU | Speed | Iter. | BLEU | Speed |
| Teacher | Transformer-big ($b = 5$) | N | 29.31 | - | N | 30.64 | - |
| *Base* | NAT drafter | 1.6 | 26.48 | **14.3×** | 1.6 | **27.16** | 14.0× |
| | AT verifier ($b = 5$) | N | 28.89 | 1.0× | N | 29.19 | 1.0× |
| | AT verifier ($b = 1$) | N | 28.73 | 1.1× | N | 29.24 | 1.1× |
| | SpecDec ($k = 25$) | 4.9 | 28.73 | 3.0× | 4.5 | 29.24 | 3.3× |
| | SpecDec++ ($k = 25$, *top-3*, $\tau = 1.0$) | 4.0 | 28.89 | 3.6× | 4.0 | 29.31 | 3.4× |
| | SpecDec++ ($k = 25$, *top-5*, $\tau = 3.0$) | 3.1 | 28.73 | **4.5×** | 3.1 | **29.52** | 4.3× |
| *Big* | NAT drafter | 1.4 | 27.35 | **15.0×** | 1.4 | **28.10** | 14.8× |
| | AT verifier ($b = 5$) | N | 29.25 | 1.0× | N | 29.74 | 1.0× |
| | AT verifier ($b = 1$) | N | 29.18 | 1.1× | N | 29.69 | 1.1× |
| | SpecDec ($k = 30$) | 4.8 | 29.18 | 3.0× | 4.2 | 29.69 | 3.3× |
| | SpecDec++ ($k = 30$, *top-3*, $\tau = 1.0$) | 4.2 | 29.32 | 3.5× | 3.8 | **29.88** | 3.6× |
| | SpecDec++ ($k = 30$, *top-5*, $\tau = 6.0$) | 2.6 | 29.15 | 5.0× | 2.5 | 29.81 | **5.1×** |

Table 10: Performance comparison between a single teacher model and teacher ensemble (3 teacher models) on WMT14 EN-DE. We report the results of both base-size and big-size SpecDec. Transformer-base/big with beam search are the speed baselines of SpecDec-base/big respectively.

---

[11]We will release all our teachers to facilitate reproducing our results.

## B.5 RESULTS FOR TRANSFORMER-12-2

Kasai et al. (2021) points out that the Transformer with a deep encoder and a shallow decoder can achieve comparable translation quality with remarkable speedups. In Table 11, we present the results of SpecDec-base with the configuration of 12 encoder layers and 2 decoder layers. The results indicate that even in the deep-shallow configuration, SpecDec/SpecDec++ can further speed up translation without quality loss.

| | Models | *E-D* | Iteration | BLEU | Speed |
|---|---|---|---|---|---|
| AT baseline | Transformer-base (w/o Seq-KD)* | 6-6 | N | 27.74 | 1.0× |
| Ours | Teacher | 6-6 | N | 29.31$^\dagger$ | - |
| | NAT drafter ($k = 25$) | 12-2 | 1.6 | 25.67 | 18.1× |
| | AT verifier ($b = 5$) | 12-2 | N | 29.13 | 1.8× |
| | AT verifier ($b = 1$) | 12-2 | N | 28.99 | 2.0× |
| | SpecDec ($k = 25$) | 12-2 | 5.6 | 28.99 | 4.6× |
| | SpecDec++ ($k = 25$, *top-3*, $\tau = 1.0$) | 12-2 | 4.7 | **29.13** | 5.0× |
| | SpecDec++ ($k = 25$, *top-5*, $\tau = 4.0$) | 12-2 | 3.4 | 29.00 | **6.3×** |

Table 11: Results of SpecDec with the deep-shallow model configuration on WMT14 EN-DE. Both the AT verifier and the NAT drafter have 12 encoder layers and 2 decoder layers. *E*: encoder depth; *D*: decoder depth. * denotes the results of AT baselines ($b = 5$) implemented by Ghazvininejad et al. (2019). $^\dagger$ indicates results obtained by Transformer-big.

## B.6 SACREBLEU AND COMET SCORES FOR WMT14 EXPERIEMENTS

We report SacreBLEU[12] (Post, 2018) and COMET[13] (Rei et al., 2020) scores in order to provide a reference for future research. SpecDec/SpecDec++ can also achieve lossless speedup even evaluated in sacreBLEU and COMET. Schmidt et al. (2022) pointed out that inconsistencies in the use of tokenized BLEU lead to deviations of up to 1.8 BLEU points. Therefore, we recommend that future research use sacreBLEU when comparing with our work.

| | BLEU | SacreBLEU | COMET |
|---|---|---|---|
| Teacher (Transformer-big) | 29.31 | 28.6 | 52.95 |
| NAT drafter ($k = 25$) | 26.48 | 26.0 | 23.63 |
| AT verifier ($b = 5$) | 28.89 | 28.2 | 51.90 |
| AT verifier ($b = 1$) | 28.73 | 28.0 | 51.53 |
| SpecDec ($k = 25$) | 28.73 | 28.0 | 51.53 |
| SpecDec++ ($k = 25$, *high-quality*) | **28.89** | **28.2** | **52.10** |
| SpecDec++ ($k = 25$, *high-efficiency*) | 28.73 | 28.0 | 51.56 |

Table 12: SacreBLEU and COMET scores on WMT14 EN-DE.

| Models) | EN→DE | DE→EN | EN→RO | RO→EN |
|---|---|---|---|---|
| Teacher (Transformer-big) | 29.31(28.6$^\dagger$) | 32.11(31.6$^\dagger$) | 34.72(33.9$^\dagger$) | 34.49(33.9$^\dagger$) |
| NAT drafter ($k = 25$) | 26.48(26.0$^\dagger$) | 30.23(29.8$^\dagger$) | 32.08(31.3$^\dagger$) | 32.21(31.7$^\dagger$) |
| AT verifier ($b = 5$) | 28.89(28.2$^\dagger$) | 32.53(32.1$^\dagger$) | 34.96(34.0$^\dagger$) | 34.86(34.2$^\dagger$) |
| AT verifier ($b = 1$) | 28.73(28.0$^\dagger$) | 32.18(31.7$^\dagger$) | 34.83(33.9$^\dagger$) | 34.65(33.9$^\dagger$) |
| SpecDec ($k = 25$) | 28.73(28.0$^\dagger$) | 32.18(31.7$^\dagger$) | 34.83(33.9$^\dagger$) | 34.65(33.9$^\dagger$) |
| SpecDec++ ($k = 25$, *high-quality*) | **28.89(28.2$^\dagger$)** | **32.56(32.1$^\dagger$)** | **35.32(34.4$^\dagger$)** | **34.98(34.3$^\dagger$)** |
| SpecDec++ ($k = 25$, *high-efficiency*) | 28.73(28.0$^\dagger$) | 32.19(31.7$^\dagger$) | 34.92(34.0$^\dagger$) | 34.80(34.2$^\dagger$) |

Table 13: BLEU and SacreBLEU (denoted by $^\dagger$) scores on WMT14 EN-DE and WMT16 EN-RO benchmarks.

---

[12] https://github.com/mjpost/sacrebleu
[13] Obtained with `wmt20-comet-da` from version `1.1.0`.

## C MEMORY ANALYSIS

### C.1 ADDITIONAL MEMORY COST BY SPECDEC

The peak memory footnote of SpecDec during inference mainly comes from three parts:

- Static AT verifier's weights

- Static NAT drafter's weights

- Intermediate variables/results

Compared with AT, the additional memory cost of SpecDec comes from the last two parts. While the static NAT drafter's weights account for the majority of the additional memory cost, the additional cost for storing intermediate variables is negligible because the NAT drafter and AT verifier decode alternatively during inference. Compared with AT, SpecDec's additional intermediate variables/results include:

- The NAT drafter's last encoder layer's representation that will not be freed until decoding finishes, which is equal to $B \cdot S \cdot d$ where $B$ is the batch size, $S$ is the sequence length and $d$ is the dimension of the model. This part is actually negligible: for example, when $B = 32$, $S = 128$, $d = 512$, this part's memory cost is only 8MB (fp32) / 4MB (fp16).

- The largest intermediate variables/results during inference:

    - For a short sequence (e.g., sentence-level inputs/outputs in MT tasks), the largest intermediate variable is the output tensor after the NAT drafter's/AT verifier's vocabulary projection layer – $B \cdot |V| \cdot k$ where $B$ is the batch size, $|V|$ is the vocabulary size and $k$ is the block size. Compared with the memory cost for storing the NAT drafter's weights, this part is usually smaller. Also $B \cdot k$ tokens can be easily divided into small batches (e.g., –softmax-batch in fairseq) for vocabulary projection to avoid massive memory cost in case $B \cdot |V| \cdot k$ is large.

    - For a long sequence (e.g., paragraph/document-level inputs/outputs in summarization tasks), the largest intermediate variable becomes the tensor for storing self-attention computation whose size increases quadratically with $S$ ($S$ is the sequence length). This variable accounts for the largest memory cost for storing intermediate results in both AT and SpecDec. Therefore, in this case, this part does not introduce additional memory cost compared with AT.

| Models | Model Weights | Batch Size | | | | |
|---|---|---|---|---|---|---|
| | | 1 | 4 | 8 | 16 | 32 |
| AT (greedy) | 232.4 | 243.2 | 271.7 | 301.4 | 366.4 | 494.6 |
| SpecDec++ | 469.8 | 483.4 | 511.8 | 551.1 | 626.0 | 774.4 |
| $\Delta$Memory | 237.4 | 240.2 | 240.1 | 249.7 | 259.6 | 279.8 |

Table 14: Peak GPU memory utilization on WMT14 EN-DE translation dataset. The results are obtained with fp32 on a single Nvidia P100 GPU. The hyperparameters of SpecDec++ are $k = 25$, *top-5*, $\tau = 3.0$.

| Models | Model Weights | Memory Cost |
|---|---|---|
| AT (greedy) | 534.6 | 696.9 |
| SpecDec++ | 1071.4 | 1246.2 |
| $\Delta$Memory | 536.8 | 549.3 |

Table 15: Peak GPU memory utilization on CNN-DM with batch size 1 with fp32 on a single Nvidia P100 GPU.

Table 14 and Table 15 show the comparisons of peak GPU memory footprint[14] (MB) between SpecDec and AT (during inference) on the above two scenarios (i.e., MT and summarization). The results are consistent with our analysis above:

> **The majority of the additional memory cost (i.e., $\Delta$Memory) is for storing the NAT drafter's weights and the additional memory cost is not very likely to significantly increase as the batch size or sequence length increases.**

Our experiments above pre-loaded both the NAT drafter and AT verifier. In fact, it is also possible to load the static weights of the AT verifier and NAT drafter in a lazy loading manner in the meantime of GPU computation to save memory as they run alternatively. However, it is usually unnecessary in practice, because for a seq2seq model deployed on modern GPUs for online service, **it is latency rather than memory that is the performance bottleneck**. See the next section for more discussion.

## C.2 MEMORY IS RARELY THE BOTTLENECK

To understand the performance bottleneck of online deployed seq2seq models, we test the latency and memory cost of T5-large[15] (around 770M parameters) with fp16 on 1 Nvidia A40 GPU running greedy decoding in the machine translation and abstractive summarization task, and show results in Table 16 and 17.

| Statistics | Batch Size | |
|---|---|---|
| | 1 | 32 |
| Latency (s) | 1.0⚠ | 1.4⚠ |
| Memory Util. (MB) | 1482 | 2003 |
| Memory Util. (%) | 3.0 | 4.0 |

Table 16: Latency and peak GPU memory utilization of T5-Large on WMT14 EN-DE.

| Statistics | Batch Size | |
|---|---|---|
| | 1 | 32 |
| Latency (s) | 2.7⚠ | 4.7⚠ |
| Memory Util. (MB) | 2999 | 7230 |
| Memory Util. (%) | 6.2 | 15.0 |

Table 17: Latency and peak GPU memory utilization of T5-Large on CNN-DM.

For MT, T5-large's latency is over 1 second which is actually too long to be accepted because most MT engines in practice require the latency to be less than 100ms. However, its memory cost is only less than 2GB – far below A40 GPU's memory capacity (i.e., 48GB[16]).

For abstractive summarization, even if the batch size increases to 32, its memory cost is still less than 50% utilization of 1 A40 GPU but its latency is already close up to 5 seconds that is too long for an online service in practice.

To sum up, we now understand latency is the bottleneck of seq2seq models for online deployment in most cases. Therefore, we do not think additional memory cost by SpecDec will undermine its practical value; instead, we think a significant lossless acceleration even at the cost of memory (i.e., time–memory trade-off) is much more meaningful than the acceleration at the cost of quality, which should be the right path that we need to pay more attention to given much memory headroom on modern GPUs.

## D PROFILING

We show the inference time cost by modules in SpecDec++ in Table 18. The current naive implementation costs a total of approximately 16% of overall time to (sequentially) encode the input for the AT and NAT model, which can be obviously optimized. Also, the NAT decoder costs more than the AT decoder because of the multi-round computation for previously decoded tokens.

---

[14]Tested with the command `torch.cuda.max_memory_allocated()`

[15]In practice, T5-large is rarely deployed for online service because it is too large and expensive to serve.

[16]It can also easily scale to 96GB with NVIDIA NVLink connection of multiple GPUs.

| Modules | Latency(ms) | Percent(%) |
|---|---|---|
| AT Encoder | 5.65 | 7.70 |
| NAT Encoder | 5.73 | 7.80 |
| NAT Decoder | 31.44 | 42.81 |
| AT Decoder | 27.33 | 37.21 |
| Others | 3.31 | 4.48 |
| Total | 73.46 | 100 |

Table 18: Profiling of SpecDec++(*base-size*, $k = 25$, *top-5*, $\tau = 3.0$) on the WMT14 EN-DE test set.

# E    SpecDec on Various Computing Devices

We also test[17] the inference efficiency of SpecDec with the batch implementation[18] on various GPUs, as shown in Figure 5. It is obvious that more powerful devices (e.g., V100/A100) can benefit SpecDec/SpecDec++ more (i.e., higher speedup).

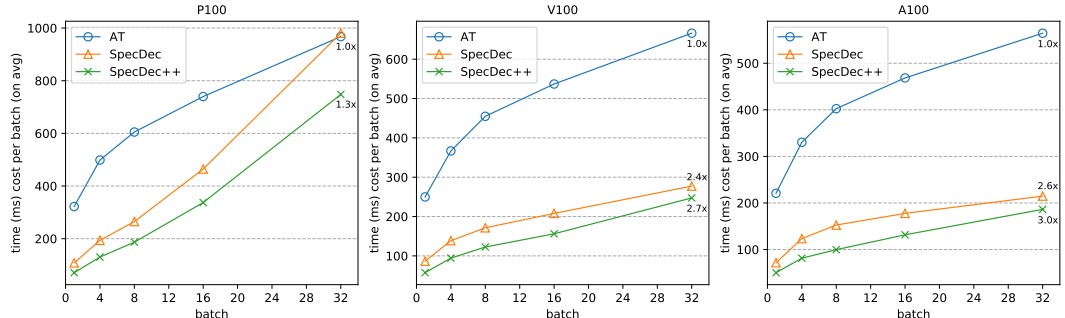

Figure 5: Inference latency of SpecDec-base ($k = 25$, *top-5*, $\tau = 3.0$) on P100 (fp32), V100 (fp16) and A100 (fp16). The results are obtained on WMT14 EN-DE. The speedup baseline ($1\times$) is AT ($b = 5$) when batch = 32.

Therefore, we believe SpecDec's speedup ratio in the future can be much higher than the number we report in this paper, because it can benefit more from evolving processor hardware that will become increasingly powerful and better at parallel computing (e.g., Nvidia H100[19] with fp8 support).

# F    Discussions of Beam Search

For possible concerns that SpecDec may not apply beam search, we make three points here:

1. As Kim & Rush (2016) mentioned, knowledge distillation largely decreases the performance gap of beam search and greedy decoding. In practice, greedy decoding can actually be comparable to beam search results after KD.

2. In practical online deployment, KD is almost used by default for enhancing the results for student models and greedy decoding is much more common than beam search because it is more cost-effective – it not only runs faster than beam search but also achieves decent performance with a student model trained through KD (as Point 1 addressed)

---

[17]We test with the batch size up to 32 because as the batch size increases, the inference latency per batch will become higher. Therefore, it is impractical to use a large batch size during inference, as Rajbhandari et al. (2022) points out.

[18]For simplifying implementation, we do not use incremental states in SpecDec to save the computation for previously decoded tokens as conventional AT, which means that its result is probably much underestimated. But even so, SpecDec/SpecDec++ still shows very promising speedup.

[19]https://www.nvidia.com/en-us/data-center/h100/

3. Beam search is also an approximate and heuristic solution, which is not a golden rule. In fact, SpecDec++ works in a similar way as beam search – it is also an approximate and heuristic solution by considering n-best and scores, which can be considered as an approximation of beam search. As shown in Table 1, it achieves comparable performance to beam search but much faster ($3\times \sim 5\times$). Therefore, we think it can replace beam search for efficient seq2seq generation.

# G  CASE STUDY

In Table 19, we represent several examples to illustrate how SpecDec/SpecDec++ generates translations. Take *Example 1* for illustration, in the first iteration, the outputs of the drafter are non-autoregressive with *multi-modality* problems like "Angaben Angaben". The verifier accepts tokens of "Nach den" and replaces the inappropriate translation "Angaben" with "vorliegenden". In the second iteration, the verification of the vanilla SpecDec finds the bifurcation at the first position thus all tokens after this position are discarded. After 4 iterations, the decoding is finished since the [EOS] token is found.

Compared with the vanilla SpecDec, in the second iteration, SpecDec++ finds that the top-$\beta$ candidate "Angaben" meets the relaxing requirement so that it accepts this token, showing that the relaxing constraints do help to accept more tokens for our proposed SpecDec model.

| | |
|---|---|
| ***Example 1- vanilla SpecDec*** | |
| SOURCE | According to the details provided , the tunnel had not yet been put into use . |
| D | Nach den Angaben Angaben war der Tunnel noch nicht in |
| V | Nach den vorliegenden ~~war war der Tunnel noch nicht in Betrieb~~ |
| D | Nach den vorliegenden Angaben war der Tunnel noch nicht in Betrieb genommen |
| V | Nach den vorliegenden Einzelheiten ~~war der Tunnel noch nicht in Betrieb genommen worden .~~ |
| D | Nach den vorliegenden Einzelheiten war der Tunnel noch nicht in Betrieb genommen [EOS] |
| V | Nach den vorliegenden Einzelheiten war der Tunnel noch nicht in Betrieb genommen worden |
| D | Nach den vorliegenden Einzelheiten war der Tunnel noch nicht in Betrieb genommen worden . [EOS] |
| V | Nach den vorliegenden Einzelheiten war der Tunnel noch nicht in Betrieb genommen worden . [EOS] |
| RESULTS | Nach den vorliegenden Einzelheiten war der Tunnel noch nicht in Betrieb genommen worden . |
| ***Example 2-vanilla SpecDec*** | |
| SOURCE | Yesterday , Gut@@ acht 's Mayor gave a clear answer to this question . |
| D | Gestern hat der Bürger@@ meister von Gut@@ acht eine klare |
| V | Gestern hat Gut@@ ~~Bürger@@ meister von Gut@@ acht eine klare Antwort~~ |
| D | Gestern hat Gut@@ acht@@ ts Bürger@@ meister eine klare Antwort auf diese Frage |
| V | Gestern hat Gut@@ ach@@ s ~~Bürger@@ meister eine klare Antwort auf diese Frage gegeben~~ |
| D | Gestern hat Gut@@ ach@@ ts Bürger@@ meister eine klare Antwort auf diese Frage gegeben |
| V | Gestern hat Gut@@ ach@@ ts Bürger@@ meister eine klare Antwort auf diese Frage gegeben . |
| D | Gestern hat Gut@@ ach@@ ts Bürger@@ meister eine klare Antwort auf diese Frage gegeben . [EOS] |
| V | Gestern hat Gut@@ ach@@ ts Bürger@@ meister eine klare Antwort auf diese Frage gegeben . [EOS] |
| RESULTS | Gestern hat Gut@@ ach@@ ts Bürger@@ meister eine klare Antwort auf diese Frage gegeben . |
| ***Example 1-SpecDec++*** | |
| SOURCE | According to the details provided , the tunnel had not yet been put into use . |
| D | Nach den Angaben Angaben war der Tunnel noch nicht in |
| V | Nach den vorliegenden ~~war war der Tunnel noch nicht in Betrieb~~ |
| D | Nach den vorliegenden Angaben war der Tunnel noch nicht in Betrieb genommen [EOS] |
| V | Nach den vorliegenden Angaben war der Tunnel noch nicht in Betrieb genommen worden |
| D | Nach den vorliegenden Angaben war der Tunnel noch nicht in Betrieb genommen worden . [EOS] |
| V | Nach den vorliegenden Angaben war der Tunnel noch nicht in Betrieb genommen worden . [EOS] |
| RESULTS | Nach den vorliegenden Angaben war der Tunnel noch nicht in Betrieb genommen worden . |
| ***Example 2-SpecDec++*** | |
| SOURCE | Yesterday , Gut@@ acht 's Mayor gave a clear answer to this question . |
| D | Gestern hat der Bürger@@ meister von Gut@@ acht eine klare |
| V | Gestern hat der Bürger@@ meister von Gut@@ acht eine klare Antwort |
| D | Gestern hat der Bürger@@ meister von Gut@@ acht eine klare Antwort auf diese Frage gegeben . [EOS] |
| V | Gestern hat der Bürger@@ meister von Gut@@ acht eine klare Antwort auf diese Frage gegeben . [EOS] |
| RESULTS | Gestern hat der Bürger@@ meister von Gut@@ acht eine klare Antwort auf diese Frage gegeben . |

Table 19: Examples from the WMT14 English-German translation task. At each iteration, *D* and *V* are the outputs of the drafter and the verifier, respectively. Tokens within red blocks are the bifurcation positions. The verification pieces after the bifurcation are annotated as strikethrough. The highlighted parts are translations of previous iterations. Tokens in blue blocks are *top-β* candidates which meet the SpecDec++ requirement. The hyperparameters are $k = 10$, *top-3*, $\tau = 1.0$. '@@' is the BPE token, e.g., Gut@@ acht → *Gutacht*. The output pieces after the [EOS] token is omitted in the table.

