# OpenReview forum: "Speculative Decoding: Lossless Speedup of Autoregressive Translation"
_ICLR.cc/2023/Conference — Submitted to ICLR 2023_

### Official Review · Reviewer_KQwP · 2022-10-21

**Confidence:** 4
**Correctness:** 3
**Technical Novelty And Significance:** 2
**Empirical Novelty And Significance:** 3
**Recommendation:** 5

**Clarity, Quality, Novelty And Reproducibility:**

The paper is very clear and the presentation quality is high. The technical novelty is limited, given the similarity with Blockwise Parallel Decoding. Empirically, the results are very promising, assuming that they can be reproduced. The authors do not disclose whether they are willing to release their code and models publicly, which would be an important contribution to reproducibility.

**Strength And Weaknesses:**

Strengths:
- The motivation of the paper is strong.
- The paper is very clear and pleasant to read.
- The experimental results are promising. In particular, achieving a 3x or more speedup while performing on par with greedy decoding and beam search in both MT and summarization is a significant achievement.
- The fact that, in addition to MT, the paper also presents results for abstractive summarization is a plus.

Weaknesses:
1. The major weakness of the paper is its very high similarity with Blockwise Parallel Decoding (Stern et al., 2018). The decoding algorithm is essentially the same, with the only significant difference relying on the fact that the proposed method uses two separate models to generate drafts and verify them. The NAT model employed in this work is more advanced, but those advancements were also brought from prior work.
2. The proposed approach works as a faster alternative to autoregressive mode-seeking algorithms (greedy decoding or beam search). Extending its principles to sampling-based approaches (top-k, nucleus, typical decoding, etc) would be an interesting contribution that is missing.
3. The claim that SpecDec offers a "3× *lossless* speedup for abstractive summarization" is an overstatement, since SpecDec performs slightly worse than the teacher model.

**Summary Of The Paper:**

The paper proposes speculative decoding, a decoding scheme for conditional NLG that aims to achieve the quality of traditional autoregressive approaches while retaining the speedup improvements of non-autoregressive approaches. The proposed framework combines a non-autoregressive transformer (NAT) to generate "draft" sequences of $k$ tokens and a standard autoregressive transformer (AT) to verify the proposed drafts. Since verification is parallelizable, this approach performs faster than usual autoregressive decoding. The experimental results reported by the authors show that this approach performs on par with greedy decoding and achieves a considerable inference speedup.

**Summary Of The Review:**

Although the experimental results are convincing, the technical contribution is quite limited, as the proposed method is essentially Blockwise Parallel Decoding enhanced with other techniques from prior work. Thus, I am not convinced that the work has enough novelty to be presented in a top venue like ICLR. However, the authors' rebuttal and/or the discussion with the remaining reviewers may persuade me.

**Update after rebuttal:**
The authors’ response has reinforced my positive opinion about the empirical relevance of their work. Nonetheless, I still think this work consists of a well-engineered refinement of prior work, with limited scientific contribution. Hence, my opinion is that this work is a better fit for an NLP-specific venue and, for this reason, I decided to keep my score.

---

> ### Author Response · Authors · 2022-11-14
> **Response to Reviewer KQwP**
>
> ***Q1: The major weakness of the paper is its very high similarity with Blockwise Parallel Decoding (Stern et al., 2018). The decoding algorithm is essentially the same, with the only significant difference relying on the fact that the proposed method uses two separate models to generate drafts and verify them. The NAT model employed in this work is more advanced, but those advancements were also brought from prior work.***
>
> As an early exploration, Blockwise Decoding [1] found that "Predict-Verify" could improve efficiency. However, [1]'s exploration is very preliminary, and it didn't discover real potential of the method: its final solution shows a 3.0x speedup but with a loss of 1.7 BLEU score, which is currently uncompetitive and even much worse than many fully-NAT approaches. As a result, its proposed idea has gradually fade out away from mainstream research of efficient seq2seq in recent years, regarded as dead-end.
>
> In contrast, we formally propose and elevate "generate-and-verify" into a paradigm -- speculative decoding, aligning with "speculative execution" in computer architecture, and systematically study it to explore its real potential by showing an around 5x speedup without quality loss compared with a strong AT baseline, which is a milestone and cannot be achieved by any previous NAT work and demonstrating it as a potentially promising paradigm for efficient seq2seq generation in future, as we mentioned in "**The meaning and significance of "lossless speedup"**".
>
> *To use an analogy, [1] is like the work showing for the first time that a vehicle can run with battery without gasolin. But as it's very preliminary: its design has many limitations and its performance is much inferior to the vehicles with gasoline, its findings were not followed by much work and almost no one believed a car with battery could really be the future for cars. In contrast, SpecDec/SpecDec++ is the work that really proposes a new paradigm to manufacture practical battery electric vehicles with many research innovations to overcome their inherent limitations, which allows them to even outperform vehicles with gasoline in practical scenarios, informing the community for the first time that battery electric vehicles can be better than gasoline vehicles and are promising to be the future for cars.* **It doesn't make any sense that such a contribution should be regarded incremental** although we admit that we stand on the shoulders of giants.
>
> We hope our explanation can help you understand these differences and value our contribution properly. We look forward to further discussion if you're still confused about our result and contribution.
>
> [1] Stern et al. 2018. Blockwise Parallel Decoding for Deep Autoregressive Models.
>
> ***Q2: The proposed approach works as a faster alternative to autoregressive mode-seeking algorithms (greedy decoding or beam search). Extending its principles to sampling-based approaches (top-k, nucleus, typical decoding, etc) would be an interesting contribution that is missing.***
>
> Thanks very much for your suggestion! Sampling-based approaches introduce more diversity into translation results, but they will negatively impact inference efficiency (please see our main results of *SpecDec++-high quality* & *SpecDec++-high efficiency* as your reference samples).
>
> Nonetheless, we also agree that extending SpecDec to sampling-based methods would be an interesting research topic and will leave it to future work.
>
> ***Q3: The claim that SpecDec offers a "3× lossless speedup for abstractive summarization" is an overstatement, since SpecDec performs slightly worse than the teacher model.***
>
> We wish the reviewer could read our paper and our rebuttal "**The meaning and significance of "lossless speedup"**" to properly understand what lossless speedup means. Our comparison baseline is never the teacher model (Transformer-big), but a base-size AT model.
>
> ***Q4: Empirically, the results are very promising, assuming that they can be reproduced. The authors do not disclose whether they are willing to release their code and models publicly, which would be an important contribution to reproducibility.***
>
> Thanks for your appreciation of our experimental results！As we noted in the **Appendix B.4** of the paper, we will release all of our codes and model checkpoints publicly for reproducing our results after this work is accepted.

---

> > ### Comment · Reviewer_KQwP · 2022-11-19
> > **Thank you for your response**
> >
> > Dear authors,
> >
> > Thank you for your careful point-by-point response. It has reinforced my positive opinion about the empirical relevance of your work. In particular, I appreciate the fact that you are releasing code and models publicly. Nonetheless, I still think this work consists of a well-engineered refinement of prior work, with limited scientific contribution. Hence, my opinion is that this work is a better fit for an NLP-specific venue and, for this reason, I decided to keep my score.

---

> > > ### Author Response · Authors · 2022-11-21
> > > **Response to Reviewer KQwP**
> > >
> > > Thanks for your response! While we don't agree with some of your points, we respect your subjective opinions. We made some changes to our response to Q1 to make it clearer and we're happy to respond to any question or concern if you still have any.

---

### Official Review · Reviewer_cfnL · 2022-10-24

**Confidence:** 4
**Correctness:** 3
**Technical Novelty And Significance:** 3
**Empirical Novelty And Significance:** 3
**Recommendation:** 6

**Clarity, Quality, Novelty And Reproducibility:**

This paper is well-written and easy to follow without too much effort. The authors provide clear examples to help the reader understand the proposed algorithm. The theoretical formalization and empirical experiment are rigorous and support the main claim. Experimental setups and hyper-parameters are clearly given and should have good reproducibility. SpecDec is inspired by speculative execution in computer architecture, but no previous work applies it for sequence generation, which is not straightforward.


**Strength And Weaknesses:**

Pros:
+ The most important contribution of this paper is that the proposed accelerated decoding algorithm is lossless. Unlike other acceleration methods for AT, such as half-precision inference, SpecDec produces the same results as greedy search. For online commercial translation systems, it is important to maintain the stability of the translation results.
+ The design of SpecDec is clever. The authors have done a good job combining the advantages of NAT and AT. AT is serial, but it is parallel when using teacher-forcing. The authors take this to verify the NAT results, which greatly improves the overall parallelism.
+ The SpecDec++ variant further speeds up the decoding by relaxing the acceptance criteria and potentially gives better results than the original.
+ The authors provide an exhaustive analysis of the proposed algorithm, including the impact of hyper-parameters, the number of tokens accepted, etc.

Cons:
- The major weakness is that SpecDec can not apply to beam search (beam size > 1). At least, it is not straightforward and not discussed by the authors. This makes "lossless" a bit overclaimed since AT model with beam search usually produces better results than greedy search.
- The NAT model causes additional memory cost, which decreases the max batch size. From this point of view, SpecDec reduces parallelism to a certain extent.

Minor:
- It would be better to complete the conditions for the probabilities in Eq 6-7 to avoid misunderstanding whether the probabilities are AT or NAT models.
- Table 1: What is the batch size?

**Summary Of The Paper:**

This paper introduces a faster decoding algorithm for autoregressive translation (AT) called Speculative Decoding (SpecDec), which achieves lossless speedup. Specifically, they combine the strengths of AT and non-autoregressive translation (NAT) and adopt a *draft* and *verify* strategy. For each step:
- The NAT first generates a fixed-length sequence of draft tokens in parallel.
- Then the AT verifies the longest acceptable prefix, reserves it, and proceeds to the next step.

SpecDec's speedup comes from a high degree of parallelism: NAT is parallel, and the AT verification process uses teacher-forcing, which is also parallel. SpecDec is lossless because the verification step uses a strict strategy to determine the longest acceptable prefix, i.e., each token in the prefix must be the top-1 choice for the AT model. As a result, SpecDec produces the identical sequence as the greedy search of the AT. They also relax the criteria for the acceptable prefix and form the variant SpecDec++, which achieves more speedup and potential improvement of translation quality. They perform empirical experiments to demonstrate the superior performance of SpecDec and detail analyses of the key components and hyper-parameters.

**Summary Of The Review:**

This paper proposes a lossless acceleration algorithm for AT that cleverly combines the features of AT and NAT. The experimental results show that the proposed method performs well in terms of efficiency and performance.

---

> ### Author Response · Authors · 2022-11-14
> **Response to Reviewer cfnL**
>
> ***Q1: Table 1: What is the batch size? The NAT model causes additional memory cost, which decreases the max batch size. From this point of view, SpecDec reduces parallelism to a certain extent.***
>
> For fair comparisons with previous NAT works, the speedups in Table 1 are all obtained by running the model with one sentence at a time on a single GPU.
>
> We also provide the comparisons of SpecDec and AT with various batch sizes (1-32) in Figure 5 of **Appendix E**. We point out that SpecDec/SpecDec++ is for (online) inference. As we replied to Reviewer GbVT in Q8, **(online) inference is different from training**: training can only consider throughput (batch size can be as large as possible for higher parallism); however, (online) inference requires a balance between throughput and latency. Although continuously increasing the batch size boosts the throughput of the overall system, the latency (per batch) will gradually increase and finally exceed users' upper limit of tolerance (e.g., 100ms). Therefore, in practice, the batch size cannot be set large (let alone max batch size that uses up memory); instead, it is usually very small during inference, as mentioned in  Appendix E and also pointed out by previous work (https://arxiv.org/pdf/2201.05596: Section 5). Therefore, **it is never a problem that SpecDec decreases max batch size because max batch size is very unlikely to be used for online inference in practice**. Please see more details in Appendix E.
>
> About the extra memory cost, we have a thorough analysis in **Appendix C** of the paper, which shows that **latency rather than memory cost is the bottleneck of online deployed seq2seq models**. Therefore, we do not think additional memory cost by SpecDec will undermine its practical value; instead, we think a significant lossless acceleration even at the cost of memory (i.e., **time–memory tradeoff**) is **much more meaningful** than the acceleration at the cost of quality (i.e., **time-quality tradeoff**). which may be the right way we need to pay more attention to given much memory headroom on modern GPUs.
>
> In a nutshell, our approach is very appropriate for efficient seq2seq generation, which utilizes the resources in the most reasonable way to optimize speed (latency) without quality loss for online inference scenarios (small batch size). **It is beyond doubt a far better way (no quality loss and reduce latency) to increase computational parallelism than monotonically increasing the batch size (harmful to latency)**.
>
> ***Q2: The major weakness is that SpecDec can not apply to beam search (beam size > 1). At least, it is not straightforward and not discussed by the authors. This makes "lossless" a bit overclaimed since AT model with beam search usually produces better results than greedy search.***
>
> We sincerely wish the reviewer could read "**The meaning and significance of "lossless speedup"**" and "**[Beam search](https://openreview.net/forum?id=H-VlwsYvVi&noteId=BTif3x5AI_z)**" to properly understand our results and significance. As we addressed in Point 3 of "**[Beam search](https://openreview.net/forum?id=H-VlwsYvVi&noteId=BTif3x5AI_z)**", we do not think it makes much sense to regard the lack of beam search as a major weakness of SpecDec/SpecDec++ because our SpecDec++ plays a similar role as beam search and can achieve comparable results as beam search but much faster. We hope that our explanations can provide you with a better understanding of our proposed approach.
>
> ***Q3: It would be better to complete the conditions for the probabilities in Eq 6-7 to avoid misunderstanding whether the probabilities are AT or NAT models.***
>
> Thanks very much for your advice! We will complete the conditions in the final manuscripts.
>
> **We sincerely wish our above answers to your questions could help you better understand our contribution and reconsider the value and significance of this work.**

---

### Official Review · Reviewer_GbVT · 2022-10-24

**Confidence:** 4
**Correctness:** 3
**Technical Novelty And Significance:** 3
**Empirical Novelty And Significance:** 3
**Recommendation:** 5

**Clarity, Quality, Novelty And Reproducibility:**

The experimental setup is described clearly. To the best of my knowledge, combining NAT and AT in this way is novel.

**Strength And Weaknesses:**

The idea is motivated well, makes intuitive sense, and is described carefully. Speed-ups without loss in quality - what's not to like?

There are, of course, caveats, and it would have been good if the paper was more explicit about its limitations.  Perhaps the most obvious one is that SpecDec can only guarantee to match AT greedy output, and it is not clear how to use AT beam search as a base. The entire architecture is significantly more complex: Training a teacher model, distilling an AT and NAT model from it, and then combine both models in this decoding scheme is more cumbersome to maintain than running simple beam search. But I think that there are also non-obvious caveats when interpreting the results in Table 1 - both in terms of translation quality and speed up.

Translation quality: AT verifier b=5 and b=1 are *very* close together in terms of BLEU. It is known that KD moves beam search and greedy search closer together, but this is closer than I would expect. I also find it odd that the "AT verifier (b=1)" - besides on En-De - consistently outperforms the "Teacher" even though the teacher has the same or bigger model size and uses beam search instead of greedy search. So to get a better sense of how to read this I would suggest to
(a) report SpecDec results with the Teacher as the AT verifier
(b) report BLEURT/COMET instead / in addition to BLEU scores
Another thing to keep in mind is that (unlike SpecDec) SpecDec++ is a model combination for which an AT+NAT ensemble would be the right baseline to compare BLEU scores with.

Speed ups: The total number of compute operations is higher for SpecDec than for vanilla greedy AT decoding, so speed-ups are due to parallelization. But better parallelization can also be achieved by simple (sentence-level) batching, which (I believe) is not used in this work. The greedy AT runtime with sentence-level batching would be a valuable baseline. Furthermore, in practice, the decoder in Transformer encoder-decoders are often replaced by recurrent models due to the time complexity of self-attention.

The paper sometimes tends to oversell the contributions. For example: "demonstrating its potential to become a de facto decoding standard in the future
for efficient and lossless seq2seq generation" is a bit much. Perhaps unintentionally, calling SpecDec "lossless" suggests the absence of search errors which is misleading.

**Summary Of The Paper:**

The decoding method called SpecDec(++) that is proposed in this paper is a combination of auto-regressive and non-autoregressive machine translation (AT & NAT). The token-by-token decoding in vanilla AT is replaced by a drafting step in which the NAT model predicts k tokens in advance, and a verification step in which the k tokens are checked against the AT decoder output. NAT predictions are discarded as soon as they are not in the top-\beta AT results. The authors demonstrate speed-ups over beam search of 4.5x without significant BLEU loss.

**Summary Of The Review:**

A reasonable idea, but the reported translation quality and speed-ups would need some work to make me trust them more. Tendency to oversell.

---

> ### Author Response · Authors · 2022-11-14
> **Response to Reviewer GbVT (Part 1)**
>
> ***Q1: The paper sometimes tends to oversell the contributions. For example: "demonstrating its potential to become a de facto decoding standard in the future for efficient and lossless seq2seq generation" is a bit much. Perhaps unintentionally, calling SpecDec "lossless" suggests the absence of search errors which is misleading.***
>
> We sincerely wish the reviewer could read "**The meaning and significance of "lossless speedup"**" to properly understand and examine the signifance of this work. If you have any other questions, please feel free to let us know.
>
> We will carefully consider the "lossless" wording you mentioned and correct it if necessary in our revised manuscript. Thanks for your suggestion!
>
> ***Q2: Perhaps the most obvious one is that SpecDec can only guarantee to match AT greedy output, and it is not clear how to use AT beam search as a base.***
>
> We addressed this in "**[Beam search](https://openreview.net/forum?id=H-VlwsYvVi&noteId=BTif3x5AI_z)**", please see above.
>
> ***Q3: The entire architecture is significantly more complex: Training a teacher model, distilling an AT and NAT model from it, and then combine both models in this decoding scheme is more cumbersome to maintain than running simple beam search. But I think that there are also non-obvious caveats when interpreting the results in Table 1 - both in terms of translation quality and speed up.***
>
> As we addressed in "**[Beam search](https://openreview.net/forum?id=H-VlwsYvVi&noteId=BTif3x5AI_z)**", most online deployed seq2seq models are distilled models. **Compared to them, SpecDec only needs to distill an extra NAT model, which doesn't introduce much additional effort compared with only distilling an AT model.** Besides, what matters most is not the cost of model preparation time, but the quality and speedup during inference. The benefit to online decoding quality and efficiency by our SpecDec/SpecDec++ is beyond doubt much more signficant than the effort for model preparation which is almost negligible.
>
> We'd like to answer any question about your understanding of our results. However, please keep in mind that model quality and speed are the most important for an online deployed seq2seq model. As you said, even if it becomes more complex to prepare and maintain, SpecDec/SpecDec++ obtains such a 3x-5x speedup at inference without quality loss -- what's not to like?
>
> ***Q4: Translation quality: AT verifier b=5 and b=1 are very close together in terms of BLEU. It is known that KD moves beam search and greedy search closer together, but this is closer than I would expect. I also find it odd that the "AT verifier (b=1)" - besides on En-De - consistently outperforms the "Teacher" even though the teacher has the same or bigger model size and uses beam search instead of greedy search.***
>
> You are absolutely right! It is an empirical observation that KD substantially decreases the performance gap between beam search and greedy decoding. In fact, it is also possible that students (even with smaller model sizes) with KD can occasionally outperform their teachers in some tasks and datasets [1, 2].
>
> > Besides, the WMT16 En$\leftrightarrow$Ro results in CMLMC [3] demonstrate similar conclusions.
>
> [1] Wang et al. 2021. MiniLMv2: Multi-Head Self-Attention Relation Distillation for Compressing Pretrained Transformers.
>
> [2] Wang et al. 2021. Selective Knowledge Distillation for Neural Machine Translation.
>
> [3] Huang et al. 2022. Improving Non-Autoregressive Translation Models without Distillation.
>
> ***Q5: report SpecDec results with the Teacher as the AT verifier***
>
> The model's performance on WMT14 EN-DE with the Teacher as the AT verifier is shown below:
>
> | Model                                | Iteration | BLEU  |    Speed    |
> | ------------------------------------ | :-------: | :---: | :---------: |
> | AT verifier (Transformer-big, b = 5) |     N     | 29.31 | 1.0$\times$ |
> | AT verifier (Transformer-big, b = 1) |     N     | 28.28 | 1.1$\times$ |
> | SpecDec (k = 25)                     |    6.1    | 28.28 | 2.4$\times$ |
> | SpecDec++ (k = 25, high-quality)     |    4.1    | 28.75 | 3.6$\times$ |
> | SpecDec++ (k = 25, high-efficiency)  |    3.0    | 28.38 | 4.5$\times$ |
>
> The results demonstrate that **SpecDec is able to achieve good quality-speedup tradeoffs even directly using the Teacher as the AT verifier**. However, we don't suggest directly using a teacher model as the verifier because the teacher model is not a distilled model; thus its greedy decoding result will not be good enough, which limits the performance of SpecDec.
>
> ***Q6: report BLEURT/COMET instead / in addition to BLEU scores***
>
> The COMET scores are provided in "**Results using more evaluation metrics**".

---

> > ### Author Response · Authors · 2022-11-14
> > **Response to Reviewer GbVT (Part 2)**
> >
> > ***Q7: Another thing to keep in mind is that (unlike SpecDec) SpecDec++ is a model combination for which an AT+NAT ensemble would be the right baseline to compare BLEU scores with.***
> >
> > Here we first clarify that the main motivation of SpecDec/SpecDec++ is to speed up inference, not ensemble. Besides, as we addressed in the conclusion of the manuscript, **SpecDec++’s introduction of an additional NAT model may play a similar role as model ensemble but its goal is to substantially speed up AT**, which is contrary to the stereotype that more models (parameters) tend to slow down inference. The vanilla ensemble goes against our optimization direction, which makes the inference slower.
> >
> > ***Q8: Speed ups: The total number of compute operations is higher for SpecDec than for vanilla greedy AT decoding, so speed-ups are due to parallelization. But better parallelization can also be achieved by simple (sentence-level) batching, which (I believe) is not used in this work. The greedy AT runtime with sentence-level batching would be a valuable baseline.***
> >
> > We provide comparisons of SpecDec and AT with various batch sizes in Figure 5 of **Appendix E**. As the results demonstrate, SpecDec/SpecDec++ also works very well in a moderate batch size setting of 1~32. Please note that **(online) inference is different from training**: training can only consider throughput (batch size can be as large as possible for higher parallelism); however, (online) inference also needs to consider latency. A large batch size (e.g., >32) will lead to an increase of latency though it improves the throughput, which is undesirable. Therefore, in practice, the batch size is usually small during inference, as also pointed out by previous work (https://arxiv.org/pdf/2201.05596: Section 5). That's why we only test on 1-32 batch size. Please see more details in Appendix E.
> >
> > In a nutshell, our approach is very appropriate for efficient seq2seq generation, which utilizes the resources in the most reasonable way to optimize speed (latency) without quality loss for online inference scenarios (small batch size). **It is beyond doubt a far better way (no quality loss and reduce latency) to increase computational parallelism than monotonically increasing the batch size (harmful to latency)**.
> >
> > ***Q9: Furthermore, in practice, the decoder in Transformer encoder-decoders are often replaced by recurrent models due to the time complexity of self-attention.***
> >
> > Thanks for your suggestions. In our early study, we tried similar approaches but found that replacing the Transformer decoder with a recurrent model resulted in worse translation quality, even worse (and slower) than the previous NAT approaches. Therefore, it is actually a weaker baseline than the NAT approaches, which is why even no previous NAT research compares with it.

---

### Official Review · Reviewer_RP8T · 2022-10-25

**Confidence:** 4
**Clarity, Quality, Novelty And Reproducibility:** 1. Comparison and evaluation are conf…
**Correctness:** 3
**Technical Novelty And Significance:** 3
**Empirical Novelty And Significance:** 3
**Recommendation:** 5

**Strength And Weaknesses:**

**Strengths:**

* The proposed idea is intuitive and easy to follow;
* The generation performance looks promising, with good quality and inference speedup;
* SpecDec generalizes to different sequence generation tasks, including translation and summarization;

**Weaknesses:**

* The idea of speculative decoding has been explored before in the form of "Predict-Verify" [1]. This work is somehow incremental. While using a separate non-autoregressive model improves translation performance, it comes at the cost of increased memory usage.
* Considering recent progress on NAT which obtains comparable and even better quality than its AT counterpart, the value of this paper becomes somehow questionable.
* Comparison and experiments should be further improved to fully convince the readers.

[1] Stern et al. 2018. Blockwise Parallel Decoding for Deep Autoregressive Models

**Summary Of The Paper:**

This paper aims at accelerating translation inference without quality degeneration. The authors extend the idea of "Predict-Verify" to "Draft-Verify" and propose SpecDec which leverages a non-autoregressive model to produce the next k tokens followed by an autoregressive model verifying these predictions in parallel. Only accepted tokens are used as outputs. The authors also propose relaxed strategies for the verification which allows more drafted tokens accepted, further gaining efficiency. Results on WMT14 En-De, WMT16 En-Ro, and CNN/Daily Mail show improved inference speed with comparable and even better generation quality.


**Summary Of The Review:**

While the paper presents some interesting results, the model is incremental to the "Predict-Verify" work and its comparison is often unfair as a large teacher model is employed. Besides, it's unclear whether the proposed SpecDec can achieve better quality-speed trade-off than recent advanced NAT models and highly optimized AT models.

---

> ### Author Response · Authors · 2022-11-14
> **Response to Reviewer RP8T (Part 1)**
>
> ***Q1: The idea of speculative decoding has been explored before in the form of "Predict-Verify". This work is somehow incremental. While using a separate non-autoregressive model improves translation performance, it comes at the cost of increased memory usage***.
>
> As you commented, Blockwise Decoding [1] found that "Predict-Verify" could improve efficiency. However, [1]'s exploration is very preliminary and it didn't discover real potential of the method: its final solution shows a 3.0x speedup but with a loss of 1.7 BLEU score, which is currently uncompetitive and even much worse than many fully-NAT approaches. As a result, its proposed idea has gradually fade out away from mainstream research of efficient seq2seq in recent years, regarded as dead-end.
>
> In contrast, we formally propose and elevate "generate-and-verify" into a paradigm -- speculative decoding, aligning with "speculative execution" in computer architecture, and systematically study it to explore its real potential by showing an around 5x speedup without quality loss compared with a strong AT baseline, which cannot be achieved by any previous NAT work and demonstrating it as a potentially promising paradigm for efficient seq2seq generation in future, as we mentioned in "**The meaning and significance of "lossless speedup"**".
>
> To use an analogy, *[1] is like the work showing for the first time that a vehicle can run with battery without gasolin. But as it's very preliminary: its design has many limitations and its performance is much inferior to the vehicles with gasoline, its findings were not followed by much work and almost no one believed a car with battery could really be the future for cars. In contrast, SpecDec/SpecDec++ is the work that really proposes a new paradigm to manufacture practical battery electric vehicles with many research innovations to overcome their inherent limitations, which allows them to even outperform vehicles with gasoline in practical scenarios, informing the community for the first time that battery electric vehicles can be better than gasoline vehicles and are promising to be the future for cars.* **It doesn't make any sense that such a contribution should be regarded incremental** although we admit that we stand on the shoulders of giants.
>
> About extra memory cost, we have a thorough analysis in **Appendix C**, showing **latency rather than memory cost is the bottleneck of online deployed seq2seq models**. Therefore, we don't think additional memory cost by SpecDec will undermine its practical value; instead, we propose that a significant lossless acceleration even at the cost of memory (i.e., **time–memory tradeoff**) is much **more meaningful than** the acceleration at the cost of quality (i.e., **time-quality tradeoff**), which may be the very right way we need to pay more attention to given much memory headroom on modern GPUs. Please refer to Appendix C.2 for more details.
>
> [1] Stern et al. 2018. Blockwise Parallel Decoding for Deep Autoregressive Models.
>
> ***Q2: Comparison and evaluation are confusing. How did you evaluate BLEU? [1] showed that inconsistencies in the use of tokenized BLEU led to deviations of up to 1.7 BLEU. Besides, directly comparing BLEU scores with previous studies is meaningless as you used a Transformer-big model as the teacher while most previous studies used the Transformer-base model.***
>
> As mentioned in our paper (in Sec 5.1), we use the same BPE tokenization and vocabulary as the work of Mask-Predict for fair comparison, which is also the most popular vocab/bpe settings in AT&NAT research, e.g., Deep-Shallow,  OAXE, and CMLMC that all use the same BPE and vocabulary. Therefore, the results are fairly comparable. Also, please note that almost all previous NAT work reports BLEU (on WMT14 En-De and WMT16 En-Ro) rather than sacreBLEU, which motivates us to use BLEU instead of sacreBLEU in our paper for comparison.
>
> However, **we agree with your suggestion of using sacreBLEU in future research**, which well **aligns with our goal to set up a proper baseline for NAT comparison** as we addressed in the [General Response](https://openreview.net/forum?id=H-VlwsYvVi&noteId=UQegm5eQZG). We report the sacreBLEU scores of SpecDec in "**Results using more evaluation metrics**" to further consolidate our empirical evaluations, please see above. We will also add the results to our paper and suggest following research use sacreBLEU.
>
> **It's a misunderstanding in comparisons between SpecDec/SpecDec++ and previous work:** We mainly compare with iterative NAT models that have high translation quality, most of which used Transformer-big as their teacher. **Please check the details mentioned in our paper (Section 5.1: Model configuration)**

---

> > ### Author Response · Authors · 2022-11-14
> > **Response to Reviewer RP8T (Part 2)**
> >
> > ***Q3: While comparing speedup numbers across papers is also meaningless [2], it's good that the authors offered the speed numbers for previous studies using their own devices. If we could trust the previous numbers, then F-VAE is significantly faster than SpecDec (16.5x vs. 4.5x) and delivers comparable results to its teacher model [3]. Also, recent progress on NAT (DA-Transformer) showed promising results. What would be the advantage of SpecDec compared to these improved NAT models?***
> >
> > Please read "**The meaning and significance of "lossless speedup"**", which includes the explanation of the comparison with DA-Transformer. For F-VAE, although it's faster, its results (33.79 BLEU for EN-RO and 34.16 for RO-EN) are still far behind strong AT baselines in our paper (34.83 for EN-RO and 34.65 for RO-EN,). In contrast, our SpecDec++ can achieve 34.92 (EN-RO) and 34.80 (RO-EN) in the same setting. **It's unreasonable to draw a conclusion that F-VAE's quality can be comparable to ours** just because F-VAE outperforms its AT teachers (33.70 for EN-RO and 34.05 for RO-EN) which are much weaker than the AT baselines should be (34.72 for EN-RO and 34.49 for RO-EN, as reported in our paper), **since outperforming an underdone AT teacher through adding extra teacher-independent tricks (e.g., CTC) doesn't mean it can also be comparable to a strong AT teacher that is well trained**.
> >
> > ***Q4: Based on [1, 2, 3], please include distant language pairs (En-Zh, En-Ja) for experiments to demonstrate SpecDec, and also report speedups on the CPU. Apart from BLEU, please also provide COMET.***
> >
> > The COMET scores are provided in "**Results using more evaluation metrics**", please see above. We will also provide the results of WMT17 En-Zh after the experiments are finished.
> >
> > As we addressed in Figure 5 of **Appendix E**, SpecDec/SpecDec++ can benefit more from evolving processor hardware (modern GPUs) that will become increasingly powerful and better at parallel computing. We will also conduct thorough experiments on the Nvidia H100 GPU in future research to further tap the great potential of SpecDec/SpecDec++.
> >
> > While CPUs which are not so good at parallel computing as GPUs are not target device our approach is proposed for, SpecDec/SpecDec++ can still achieve a 1.7x~2.6x lossless speedup on Intel(R) Xeon(R) Platinum 8358P CPU @ 2.60GHz.
> >
> > ***Q5: As far as I know, semi-autoregressive models already achieve >3x speedups on CNN/Daily Mail with little quality loss.***
> >
> > IBDecoder [1] is exactly the kind of previous work we mentioned in "**The meaning and significance of "lossless speedup"**" which compares their distilled model with a weak AT model without knowledge distillation in CNN-DM. Please check its baseline results and compare with ours as follows to properly understand the differences:
> >
> > | Models        | Proposed Model |     Teacher      | AT Baseline | AT Teacher | Gap $\downarrow$ | Gap w/ our AT $\downarrow$ |
> > | :------------ | :------------: | :--------------: | :---------: | :--------: | :--------------: | :------------------------: |
> > | IBDecoder [1] |     37.03      | ??? |    36.88    |     None      |      -0.15       |           +2.93            |
> > | SpecDec       |   **39.96**    |     BART [2]     |  **39.96**  |    BART    |      +0.00       |         **+0.00**          |
> >
> > > The reported results are Rouge-L scores obtained on CNN-DM. *"???"* means that we cannot find the detailed model configuration of the teacher in the paper.
> >
> > [1] Zhang et al. 2020. Fast Interleaved Bidirectional Sequence Generation.
> >
> > [2] Lewis et al. 2020. BART: Denoising Sequence-to-Sequence Pre-training for Natural Language Generation, Translation, and Comprehension.

---

### Author Response · Authors · 2022-11-14
**Author's Response to all reviewers (Part 1)**

We thank the reviewers for their valuable suggestions and deeply appreciate the positive comments that our work is interesting and well-motivated, showing promising results with good quality-speedup tradeoffs, and is in the high quality of writing.

In the following part of our response, we will refer to reviews in the order as they are shown on openreview: R1 = RP8T, R2 = GbVT, R3 = cfnL, R4 = KQwP.

**Response to some general questions**:

***Answers to a number of reviewers' questions can be found in the Appendix that we carefully prepared in advance.***  For example, **Appendix C** answers the concern regarding the extra memory cost (R1, R3), **Appendix E** shows the speedup effects at various batch sizes (R2) and **Appendix B** show detailed results and our willingness to release all our codes and models for reproducibility (R4). We would really appreciate it if you could take a look at the Appendix before reading the replies, which should resolve most of your concerns.

As for the reviewers’ remaining questions, we here first highlight several key clarifications and additional experimental results to respond to general questions before answering detailed questions of each reviewer.

**1.The meaning and significance of "lossless speedup" (To R1, R2, R3, R4)**

**Meaning:** In this paper’s base-size model setting, "lossless speedup" means that our base-size SpecDec/SpecDec++ achieve the same or better results than greedy decoding of the **base-size autoregressive model** besides the acceleration of decoding speed.

**Significance:** While we appreciate valuable feedback from reviewers, it is beyond our expectation that some reviewers consider such a 3x~5x speedup without quality loss compared with autoregressive (greedy) decoding nothing remarkable. This is actually **a very big misconception** leading to an underestimation of our contributions, which may arise from some one-sided conclusions/claims from some previous work that NAT was already comparable to or even better than AT. In fact, however, **what most previous NAT papers compare with are weak AT baselines that are underdone**: either they compare their proposed models with a weak AT model without knowledge distillation (but their proposed models are distilled from a powerful teacher), or their reported AT scores tend to be lower than they should be if the AT baselines are well trained. It was totally understandable and less unblameable for early NAT research (2017-2020) to underreport or underdo AT baselines because of a huge gap between NAT and AT that time. But now, we should not be misled by those numbers: In fact, **there was still a large gap of performance between NAT and AT (even greedy decoding) before this work if the AT baseline could be as carefully established as those proposed NAT models**. Unfortunately, most previous NAT papers do not have the motivation to develop a strong AT baseline, see the following table:

| Models             |  Proposed Model  |  Teacher  | AT Baseline | AT Teacher |    Imple.    | Gap $\downarrow$ | Gap w/ our AT $\downarrow$ | Speedup  |
| :----------------- | :-------: | :-------: | :----------: | :--------------: | :------------------------: | :------: | :----------------: | :----------------: |
| CMLM [1]           |   27.03   |   big   |   27.86   |   big   | $\checkmark$ |      +0.83       |           +1.70            |   1.7x   |
| DA-Transformer [2] |   27.91   |   big   |   28.54   |   big   | $\checkmark$ |      +0.63       |           +0.82            |   7.0x   |
| RewriteNAT [3] | 27.83 | big | 27.82 | none |      -       |      -0.01       | +0.90 | 2.8x |
| Imputer [4] | 28.20 | big | 27.80 | none |      -       |      -0.40       | +0.53 | 3.9x |
| CMLMC [5]         |   28.37   |   big   |   28.30   |   big   |   $\times$   |      -0.07       |           +0.36            |   1.5x   |
| SUNDAE [6]        |   28.46   |   big   |     27.30     |     none     |      -       |        -1.16        |           +0.27            |   1.4x   |
| SpecDec            | **28.73** | big | **28.73** | big | $\checkmark$ |      +0.00       |         **+0.00**          | **3.0x** |
| SpecDec++          | **28.89** | big | **28.73** | big | $\checkmark$ |    -0.16     |         **-0.16**          | **4.5x** |

> The results are obtained on WMT14 EN-DE. *”Teacher"* is the teacher model for distilling the proposed model; while *”AT Teacher"* means the teacher for distilling the AT baseline (*"none"* means the AT baseline is trained from scratch without KD). *"Imple."* means that the results of AT baseline with KD shown in the papers are based on their own implementation. **Except DA-Transformer whose AT baseline (28.54 BLEU) is close to ours (28.73 BLEU), others' AT baselines are all significantly lower than our AT baseline (with greedy decoding)**.

---

> ### Author Response · Authors · 2022-11-14
> **Author's Response to all reviewers (Part 2)**
>
> The above table shows in the big (teacher)->base (student) distillation setting, although some previous papers (e.g., Imputer, CMLMC and RewriteNAT) outperform their AT baselines (i.e., Gap<0) and claim their approaches are comparable to AT, their AT baselines are all weak -- **lagging 0.3~1.4 BLEU behind our implemented AT baseline** (greedy decoding) as reported in our paper. In fact, **AT (even using greedy decoding) is a super strong baseline** which was clearly underdone/underreported by many previous papers, and **it can be far better than previous NAT work as long as it's carefully established**. With this fact in mind, it is not hard to understand why **our contribution is significant and profound**: SpecDec/SpecDec++ is **the only approach** to really achieve exactly the same and even better result than the strong AT baseline with significant 3x-5x speedup for the first time (in contrast to 1.4x-1.5x speedup with quality loss by CMLMC and SUNDAE) in the standard seq2seq benchmarks (WMT and CNN-DM). There's even no alternative (excluding optimization orthogonal to ours, like quantization) that can achieve similar results so far. Moreover, we present an insight that moderately investing in an auxiliary model (NAT drafter) for speculation can result in a large speedup without quality loss. And **we elevate this insight into an inspiring paradigm "speculative decoding", aligning with "speculative execution" in computer architecture**, to which the NLP (even machine learning) community pays very little attention.
>
> More importantly, we only use simple models/methods and under-optimized implementation in this preliminary work. As mentioned in our paper, **our proposed speculative decoding paradigm has much headroom for its speed optimization** and it's promising to be a practical solution to efficient seq2seq generation in the future. Given the promising paradigm with strong results, significant advancement and wide application scenarios for efficient seq2seq generation in practice,  **we don’t think we're overselling this work**. On the contrary, it's some previous NAT papers saying their proposed NAT methods are comparable to AT (but their underdone AT baselines are much weaker than they should be) that oversell/overclaim their merits and results, **more or less misleading reviewers into undervaluing our work** which achieves a milestone to truly match AT's results with a 3x~5x speedup. We hope our above explanation can help resolve the misconception. We believe reviewers are capable of distinguishing our paper from those really overclaiming and can reevaluate our contributions after properly examining and understanding its significance. **If reviewers are still confused about our results or contribution, please feel free to leave a message. We'll be always happy to answer and clarify**.
>
> Moreover, although it's common that baselines are underdone/unreported in research papers, given that the misconception may lead to a misjudgment and potentially impede progress in the research problem, **we're increasingly aware of the necessity to set up a proper baseline for the research of efficient seq2seq generation**, which not only can avoid misconception for the audience (readers) but also set up a higher standard for following research papers to prevent them enjoying beating weak AT baselines. For this goal, **we'd like to release our codes and checkpoints for reproducibility to advance research in this problem.**
>
> [1] Mask-Predict: Parallel Decoding of Conditional Masked Language Models.
>
> [2] Directed Acyclic Transformer for Non-Autoregressive Machine Translation.
>
> [3] Learning to Rewrite for Non-Autoregressive Neural Machine Translation.
>
> [4] Non-Autoregressive Machine Translation with Latent Alignments.
>
> [5] Improving Non-Autoregressive Translation Models without Distillation.
>
> [6] Step-unrolled Denoising Autoencoders for Text Generation.
>
> **2.Results using more evaluation metrics (To R1, R2)**
>
> We accept the suggestion by additionally reporting sacreBLEU and COMET on WMT14 EN-DE as follows:
>
> | Model                               | BLEU  | sacreBLEU |   COMET   |
> | ----------------------------------- | :---: | :-------: | :-------: |
> | Teacher                             | 29.31 |   28.6    |   52.95   |
> | NAT drafter (k=25)                  | 26.48 |   26.0    |   23.63   |
> | AT verifier (b = 5)                 | 28.89 |   28.2    |   51.90   |
> | AT verifier (b = 1)                 | 28.73 |   28.0    |   51.53   |
> | SpecDec (k = 25)                    | 28.73 |   28.0    |   51.53   |
> | SpecDec++ (k = 25, high-quality)    | 28.89 |   28.2    | **52.10** |
> | SpecDec++ (k = 25, high-efficiency) | 28.73 |   28.0    |   51.56   |
>
> > The COMET scores are obtained with wmt20-comet-da from version 1.1.0.
>
> It's clear that SpecDec/SpecDec++ consistently achieves lossless speedup even evaluated in sacreBLEU and COMET.

---

> > ### Author Response · Authors · 2022-11-14
> > **Author's Response to all reviewers (Part 3)**
> >
> > **3.Beam search (To R2, R3)**
> >
> > Regarding the reviewer's concern that SpecDec may not apply beam search, we make three points here:
> >
> > 1. As reviewer GbVT mentioned, knowledge distillation largely decreases the performance gap of beam search and greedy decoding. In practice, greedy decoding can actually be comparable to beam search results after KD.
> >
> > 2. In practical online deployment, **KD is almost used by default for enhancing the results for student models** and **greedy decoding is much more common than beam search** because it is more cost-effective -- it not only runs faster than beam search but also achieves decent performance with a student model trained through KD (as Point 1 addressed)
> >
> > 3. Beam search is never a golden rule (just as a fuel tank is not indispensable in a vehicle) for seq2seq models; instead, it is only an alternative to decoding with approximation and heuristics. **In fact, SpecDec++ works in a similar way as beam search -- it is also an approximate and heuristic solution by considering n-best and scores.** Given that SpecDec++ can achieve comparable performance to beam search but much faster (with about 3x~5x speedup), we don't really need the exact beam search. **Therefore, it makes little sense to regard a lack of beam search implementation in SpecDec/SpecDec++ as a major weakness**, just as is unreasonable to criticize a battery electric vehicle having no fuel tank.

---

### Author Response · Authors · 2022-11-19
**List of Revisions**

Dear Reviewers,

We thank all the reviewers for their work and their valuable comments. We have uploaded a new revision to the manuscript, in which we incorporate several comments from the reviewers. Some major revisions are:

1. Adding COMET scores on WMT14 EN$\rightarrow$DE and sacreBLEU scores on WMT14 EN$\leftrightarrow$DE and WMT16 EN$\leftrightarrow$RO in Appendix B.6.
2. Adding discussions of beam search in Appendix F.
3. Clarifying some details and fixing minor issues, e.g., the conditions (6) and (7) in Sec 4.

We hope the reviewers have another look at the paper. **We sincerely hope the reviewers acknowledge and respond to our responses so that we can address further concerns (if any) within the rebuttal time period**.

We deeply appreciate your time and effort in reviewing our paper.

Best regards

---

### Decision · Program_Chairs · 2023-01-20

**Decision:**

Reject

**Justification For Why Not Higher Score:**

While the approach is reasonable, the tendency to oversell the work makes it hard to recommend for acceptance. The technical contribution is limited compared to Stern et al (2018), further comparison with beam search is necessary, and claims about the practical significance of the results do not discuss the possibility of batched greedy decoding.


**Justification For Why Not Lower Score:**

n/a

**Metareview: Summary, Strengths And Weaknesses:**

The submission introduces a method for improving the efficiency of machine translation models, by first drafting candidate next tokens with a non-autoregressive model, and then verifying them with an auto-regressive model.

The work is a clever and interesting approach that substantially improves the efficiency of auto-regressive decoding, while retrieving exactly the same output as greedy decoding. The paper also includes a nice extension can slightly improve performance compared to greedy decoding. The idea may well be useful in practice.

However, reviewers raised several important concerns, generally reflecting a tendency to oversell the work. The technical approach is closely related to [1] - which is ok, except that the positioning of the submission overplays its novelty. Further, the comparison to beam search decoding is weak. When claiming lossless decoding compared, beam search is a natural comparison, and results should be included in the main results tables - I disagree with the authors that SpecDec++ is similar to beam search. Further, while the authors make many claims around the practical implications of their approach, their method adds a large amount of complexity compared to greedy decoding, and for many practical scenarios then batched greedy decoding may be a more appropriate comparison. Describing the submission as "significant and profound" in the author response does little to address the concerns around overselling.

In communication to the AC, the authors are highly critical of the review quality. On reading the reviews and submission, I think that this criticism is unwarranted, and that the reviews reflect careful, good faith attempts to fairly evaluate the submission. To the extent that concerns raised in the reviews are due to misunderstandings, the authors should look to improve the clarity of their presentation.

[1]Stern et al. 2018.Blockwise Parallel Decoding for Deep Autoregressive Models



**Summary Of Ac-Reviewer Meeting:**

n/a